# Recruitment of autophagy initiator TAX1BP1 advances aggrephagy from cargo collection to sequestration

Bernd Bauer [iD][1,2,3], Jonas Idinger[1,2], Martina Schuschnig[1,2], Luca Ferrari[1,2] & Sascha Martens [iD][1,2 ✉]

## Abstract

Autophagy mediates the degradation of harmful material within lysosomes. In aggrephagy, the pathway mediating the degradation of aggregated, ubiquitinated proteins, this cargo material is collected in larger condensates prior to its sequestration by autophagosomes. In this process, the autophagic cargo receptors SQSTM1/p62 and NBR1 drive cargo condensation, while TAX1BP1, which binds to NBR1, recruits the autophagy machinery to facilitate autophagosome biogenesis at the condensates. The mechanistic basis for the TAX1BP1-mediated switch from cargo collection to its sequestration is unclear. Here we show that TAX1BP1 is not a constitutive component of the condensates. Its recruitment correlates with the induction of autophagosome biogenesis. TAX1BP1 is sufficient to recruit the TBK1 kinase via the SINTBAD adapter. We define the NBR1–TAX1BP1-binding site, which is adjacent to the GABARAP/LC3 interaction site, and demonstrate that the recruitment of TAX1BP1 to cargo mimetics can be enhanced by an increased ubiquitin load. Our study suggests that autophagosome biogenesis is initiated once sufficient cargo is collected in the condensates.

**Keywords** Quality Control; Selective Autophagy; Aggrephagy p62; NBR1; TAX1BP1
**Subject Category** Autophagy & Cell Death

## Introduction

Maintaining a healthy proteome is essential for cellular homeostasis and the survival of the organism. Cells have evolved numerous overlapping pathways to ensure that misfolded proteins are either refolded or swiftly ubiquitinated and removed by degradation (Chen et al, 2011). Apart from the ubiquitin-proteasome system, macroautophagy (hereafter autophagy) is a major pathway for the degradation of misfolded, ubiquitinated proteins (Jayaraj et al,

2020). The selective removal of these proteins is termed aggrephagy (Lamark and Johansen, 2021). In aggrephagy, cargo proteins are collected in larger condensates and subsequently sequestered within autophagosomes, which form de novo around the condensates by the concerted action of the autophagy machinery (Bauer et al, 2023; Melia et al, 2020; Nishimura and Tooze, 2020). Defects in this process are associated with various neurodegenerative diseases (Yamamoto et al, 2023).

Multiple pathways of aggrephagy and the sequestration of protein condensates in autophagosomes have been described in various species. For example, in *S. cerevisiae* liquid-like condensates composed of prApe1 are delivered into the vacuole by a pathway termed cytoplasm-to-vacuole targeting (Cvt) (Yamasaki et al, 2020), while in C. elegans gel-like PGL granules are delivered into lysosomes by autophagy (Zhang et al, 2018). In mammalian cells proteins can be sequestered in dynamic condensates marked by the p62 protein before being targeted by autophagy (Agudo-Canalejo et al, 2021; Sun et al, 2018; Zaffagnini et al, 2018) and solid aggregates have been reported to be delivered for autophagy by the CCT2 protein (Ma et al, 2022).

Cargo receptors orchestrate selective autophagy by binding the cargo material, recruiting the autophagy machinery for autophagosome biogenesis and subsequently linking the cargo to the GABARAP and LC3 decorated autophagosomal membrane (Adriaenssens et al, 2022). At least three cargo receptors cooperate in the degradation of ubiquitinated proteins by aggrephagy (Fig. 1A). p62 forms polymers and drives the formation of condensates with ubiquitinated cargoes through its UBA-mediated interaction with the ubiquitin chains (Bjorkoy et al, 2005; Ciuffa et al, 2015; Kageyama et al, 2021; Sun et al, 2018; Turco et al, 2021; Zaffagnini et al, 2018). NBR1 binds to p62 via its N-terminal PB1 domain and assists p62 in cargo condensation by contributing a high affinity UBA domain to the complex (Kirkin et al, 2009; Lamark et al, 2003; Turco et al, 2021). Additionally, NBR1 mediates the recruitment of TAX1BP1 to the condensates (Turco et al, 2021), which in turn is the main recruiter of the core autophagy protein FIP200 (Turco et al, 2021). A complex network of interactions then mediates the assembly and activation of the autophagy machinery including the TBK1 kinase at the p62-ubiquitin condensates (Feng et al, 2023; Schlütermann et al, 2021).

[1]Max Perutz Labs, Vienna Biocenter Campus (VBC), Dr.-Bohr-Gasse 9, 1030 Vienna, Austria. [2]University of Vienna, Max Perutz Labs, Department of Biochemistry and Cell Biology, Dr.-Bohr-Gasse 9, 1030 Vienna, Austria. [3]Vienna Biocenter PhD Program, a Doctoral School of the University of Vienna and the Medical, University of Vienna, A-1030 Vienna, Austria. ✉E-mail: sascha.martens@univie.ac.at

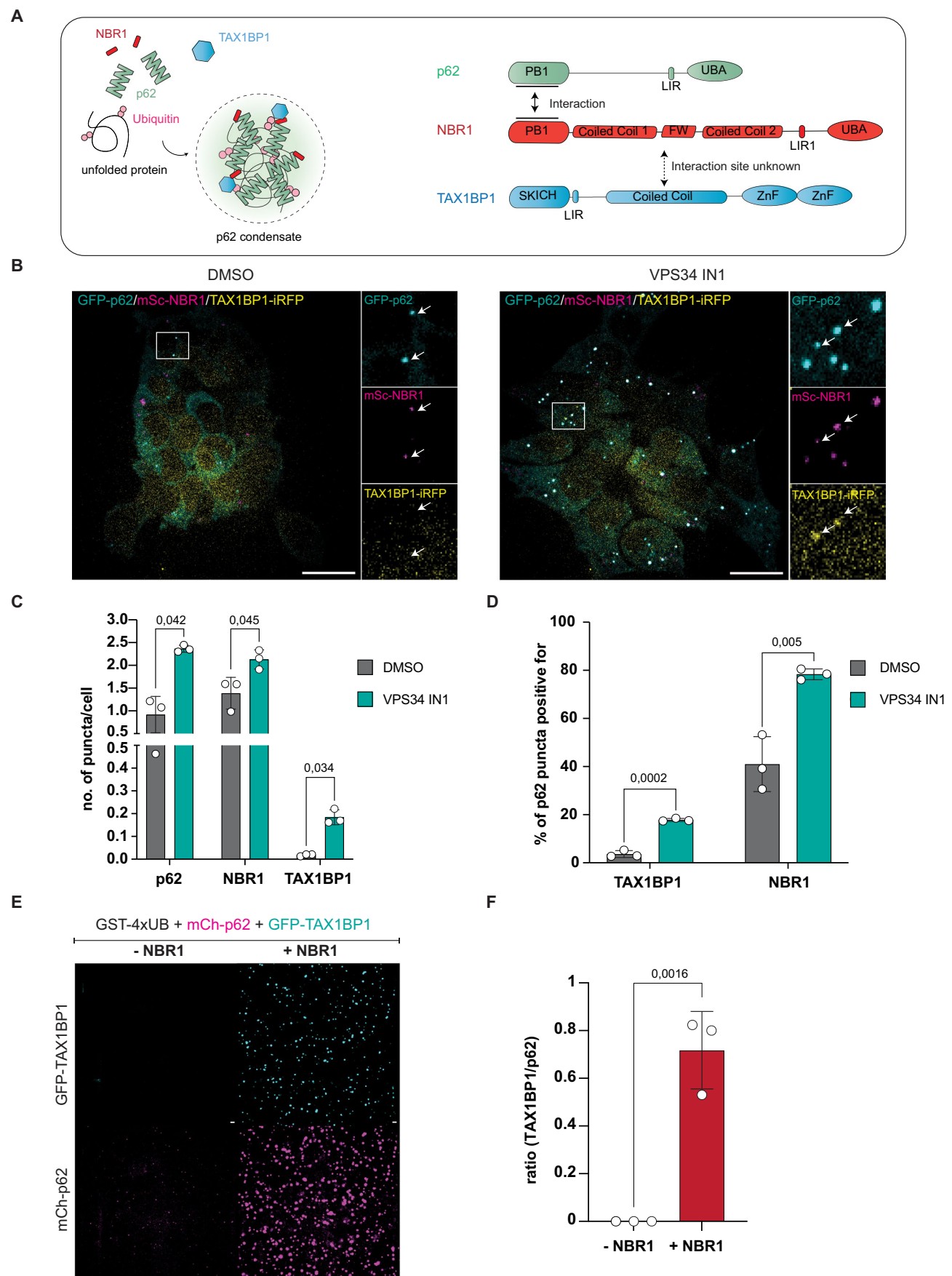

**Figure 1. TAX1BP1 is not a constitutive component of p62 condensates in cells.**

(A) Scheme of the three main cargo receptors in aggrephagy: p62, NBR1, and TAX1BP1. (B) Representative live-cell images of HAP1 cells expressing endogenously tagged p62, NBR1, and TAX1BP1 untreated and upon autophagy inhibition by VPS34 IN1 (scale bar = 20 μm). p62 & NBR1 form overlapping puncta in basal conditions and TAX1BP1 forms puncta together with them upon addition of VPS34 IN1 (white arrows). (C) Quantification of the number of puncta of the different cargo receptors forming with and without autophagy inhibition in (B). Puncta were counted from a single focal plane. (D) Quantification of the degree of overlap between the cargo receptors in (A) on a single focal plane. (E) Representative image of the in vitro condensation assay +/− NBR1 (scale bar = 10 μm). (F) Quantification of the experiment in (E). Ratio between TAX1BP1 and p62 condensates at 15 min after addition of GST-4xUB is shown. Data in (C–E) show the mean +/− s.d. from three independent experiments. Unpaired *t* tests were performed in (B, C, E). Source data are available online for this figure.

Depletion of TAX1BP1 impedes the clearance of proteins aggregates in cells and mutations in this protein lead to the accumulation of ubiquitinated proteins in the brain (Sarraf et al, 2020).

The recruitment of TAX1BP1 to the p62-ubiquitin condensates is thus a crucial step during aggrephagy and its recruitment may trigger the switch from cargo collection in the condensates to condensate sequestration by autophagosomes. However, the mechanistic basis for the recruitment of TAX1BP1 to the condensates is still enigmatic. Through a combination of biochemical reconstitution and cell biology approaches, we reveal that TAX1BP1 is not a constitutive component of p62 condensates in cells. Dissection of the molecular basis for the interaction between TAX1BP1 and NBR1 revealed that GABARAP proteins compete with TAX1BP1 for NBR1 binding and decrease TAX1BP1 recruitment. Since ubiquitin binding promotes the recruitment of TAX1BP1 to p62-ubiquitin condensates, we propose a model in which TAX1BP1 acts as a ubiquitin sensor and is only recruited to p62 condensates once a certain local ubiquitin concentration threshold is exceeded.

# Results

## TAX1BP1 is not a constitutive component of p62 condensates in cells

Since TAX1BP1 is a crucial factor for the initiation of autophagosome biogenesis at the p62 condensates, we wanted to follow its dynamics in relation to p62 and NBR1 in cells. To this end, we endogenously tagged p62 with GFP, NBR1 with mScarlet and TAX1BP1 with iRFP. Comparison of the autophagic flux between the engineered and parental cell line revealed no significant difference in the stabilization of TAX1BP1, NBR1, p62 as well as the activated form of p62 phosphorylated at S349 by Bafilomycin (Baf) treatment (Fig. EV1A,B). Under basal conditions p62 and NBR1 colocalized extensively when imaged by spinning disk live-cell microscopy (Fig. 1B, left panel). By contrast, TAX1BP1 showed an even distribution throughout the cell and a low level of colocalization with p62 and NBR1 (Fig. 1B, left panel). However, treatment with the class III phosphatidylinositol 3-phosphate kinase (PI3KC3) inhibitor VPS34 IN1 (Bago et al, 2014), which blocks autophagosome biogenesis, induced foci formation for TAX1BP1 (Fig. 1B, right panel). These foci colocalized with a fraction of the p62 and NBR1 condensates (Fig. 1C,D). Consistently, TAX1BP1 also accumulated at p62 condensates in ATG14 KO, ATG7 KO and FIP200 KO cells (Fig. EV1C). The finding that TAX1BP1 is not a constitutive component of the p62 condensates

and that we needed to arrest autophagosome biogenesis to detect robust colocalization of TAX1BP1 with these condensates was surprising because in a fully reconstituted system in which condensate formation is induced by mixing GST-4xUB, p62, and NBR1, TAX1BP1 is recruited to almost all condensates in a NBR1 dependent manner (Fig. 1E,F) (Turco et al, 2021; Zaffagnini et al, 2018). These results suggested that there is an unknown level of regulation for the recruitment of TAX1BP1 to the p62 condensates in cells. Our data further suggested that the recruitment of TAX1BP1 may coincide with induction of autophagosome biogenesis and thus the switch from cargo collection to its sequestration by autophagosomes.

## TAX1BP1 is crucial for autophagic flux of p62 condensates

To confirm the role of TAX1BP1 as a regulator of autophagic flux of p62 condensates in our cell system, we generated a TAX1BP1 knockout (KO) in the background of a GFP-p62 expressing cell line. Employing live-cell imaging, we found that the overall size and total number of p62 condensates was significantly increased in the TAX1BP1 KO cells compared to the parental cell line (Fig. 2A,B). The addition of the autophagy inhibitor VPS34 IN1 did not lead to a significant further increase in the number of p62 condensates in the TAX1BP1 KO cells, unlike in the parental cell line (Fig. 2B). Consistently, NBR1 levels were significantly increased in the TAX1BP1 KO cell line (Fig. 2C,D). Furthermore, TAX1BP1 KO cells showed a robust increase in the number of p62 condensates compared to the parental cell line upon inhibition of the proteasome by MG132 (Appendix Fig. S1A). Taken together, our data show that autophagic flux of p62 condensates is regulated by TAX1BP1 in these cells. Next, we probed for colocalization between p62 and the autophagy machinery by immunofluorescence. This revealed a significant reduction in colocalization between p62 condensates and the autophagy core components ATG9A and FIP200 in the absence of TAX1BP1 (Fig. 2E,F; Appendix Fig. S1B) (Turco et al, 2021), supporting the hypothesis that TAX1BP1 is the main recruiter of the autophagy machinery.

## TAX1BP1 can recruit TBK1 to p62 condensates via the adapter proteins SINTBAD/NAP1

To assess whether TAX1BP1 also mediates the recruitment of other components of the selective autophagy machinery to the condensates, we focused on TBK1. The TBK1 kinase has emerged as a major pro-autophagic factor in selective autophagy and binds to TAX1BP1 via the NAP1/SINTBAD adapters (Fu et al, 2018). In addition, it has recently been shown that the depletion of TAX1BP1

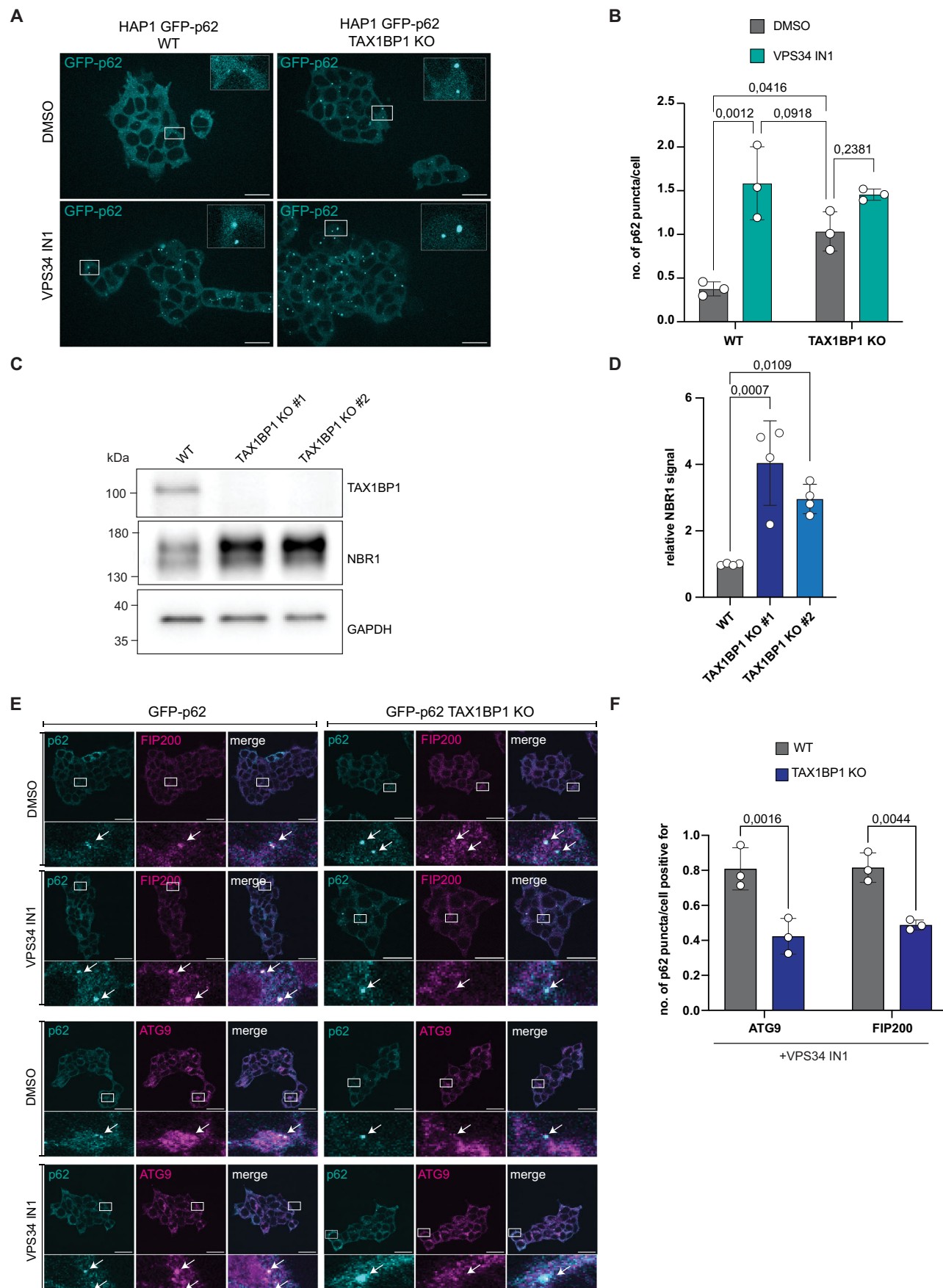

**Figure 2.  TAX1BP1 is crucial for autophagic flux of p62 condensates in cells.**

(A) Representative live-cell images of GFP-p62 expressing WT and TAX1BP1 KO cells untreated and upon VPS34 IN1 treatment (scale bar = 20 μm). (B) Quantification of the number of p62 puncta per cell on a single focal plane in (A). (C) Western blot of NBR1 for the WT and TAX1BP1 KO cells. (D) Quantification of the western blot in (C). NBR1 protein levels were quantified as a relative to WT. (E) Representative immunofluorescence images of the WT and TAX1BP1 KO cells untreated and upon autophagy inhibition stained for p62 and either FIP200 or ATG9 (scale bar = 20 μm). FIP200 and ATG9 colocalization with p62 puncta was abrogated in TAX1BP1 KO cell line (white arrows). (F) Quantification of images in (E). Quantification was performed using VPS34 IN1 samples. Puncta were counted from a single focal plane. Data in (B, F) show the mean +/− s.d. from three independent experiments. Data in (D) shows the mean +/− s.d. from four independent experiments. Two-way ANOVA (analysis of variance) with Sidak's multiple comparison test was performed in (B, F). One-way ANOVA with Dunnett's multiple comparison test was performed in (D). Source data are available online for this figure.

results in reduced TBK1 activation under basal conditions (Yamano et al, 2024). Indeed, we found that phosphorylation of p62 at S403, which is mediated by TBK1 (Matsumoto et al, 2011; Pilli et al, 2012), was abolished in the TAX1BP1 KO cells. This was particularly evident upon inhibition of autophagy with VPS34 IN1, which resulted in the accumulation of phosphorylated p62 (Fig. 3A,B). We also observed a significant reduction in phosphorylation of the TBK1 adapter protein NAP1 (Fig. 3A,C). Thus, we speculated that TAX1BP1 might promote the recruitment of TBK1 to p62 condensates. Consistently, we found that NAP1 showed a significant reduction in colocalization with p62 condensates in the TAX1BP1 KO cells compared to the parental cell line (Fig. EV2A,B). In addition, the colocalization of p62 condensates with active TBK1, phosphorylated at S172, was also reduced in the TAX1BP1 KO cells (Figs. 3D,E and EV2C). Since the TBK1 adapters NAP1 and SINTBAD are also required for the recruitment of TBK1 to p62 condensates (Fig. EV2D), we speculated that TAX1BP1 recruits TBK1 to p62 condensates via these adapter proteins. To test this more directly, we reconstituted these steps in vitro. We incubated p62, NBR1, TAX1BP1, the TBK1 adapters SINTBAD or NAP1 as well as TBK1. The addition of GST-4xUB induced p62-driven condensation and revealed that TAX1BP1 is both necessary and sufficient to recruit TBK1 to p62 condensates via the NAP1/SINTBAD adapters (Figs. 3F–J and EV2E–H). SINTBAD and TBK1 appeared to be equally distributed within the p62 condensates, in contrast to TAX1BP1, which showed enrichment at the center of the condensates in vitro (Fig. EV2F). Taken together, our data showed that TAX1BP1 plays a major role in mediating TBK1 recruitment to p62 condensates and thus potentially promotes the local phosphorylation of p62 and other components in these structures (Fig. 3K).

## TAX1BP1 binds to the LIR motif of NBR1 via its N-domain in vitro

How TAX1BP1 is recruited to the p62 condensates is not well understood. While it was shown that the interaction with NBR1 requires a central part of TAX1BP1, referred to as N-domain (aa420-506) (Ohnstad et al, 2020), the mechanistic basis of this interaction remains unexplored. We previously showed that the FW domain of NBR1 is sufficient for the interaction with TAX1BP1 (Turco et al, 2021). However, the FW domain is not essential because its deletion did not abolish the binding to TAX1BP1. We therefore set out to systematically define the interaction between TAX1BP1 and NBR1 with the aim to reveal the basis of a potential regulatory mechanism. We started by testing if the TAX1BP1 N-domain is not only required but also sufficient for the interaction

with NBR1. Employing a microscopy-based protein–protein interaction assay we found that the N-domain is indeed sufficient to recruit NBR1 (Figs. 4A,B and EV3A). Next, we determined the corresponding binding region in NBR1. Multiple attempts to define this interaction site using AlphaFold 2 gave inconclusive results. We therefore used a peptide array where we spotted NBR1 peptides on a membrane. Each peptide consisted of 12 amino acids with two residues overlap between peptides. We then probed for interacting peptides with recombinant TAX1BP1. We identified two regions capable of binding to TAX1BP1: parts of the FW domain, previously implicated to bind TAX1BP1 weakly (Turco et al, 2021) and the LC3-interacting region (LIR) 1 (Kirkin et al, 2009) (designated as binding region 1 and 2, respectively; aa347–aa365 and aa731–aa749) (Fig. 4C). Next, we tested if the deletion of the identified binding regions would abrogate the binding between NBR1 and TAX1BP1. Transfection of HeLa PentaKO cells (lacking p62, NBR1, TAX1BP1, NDP52, and OPTN (Lazarou et al, 2015)) with corresponding deletions of NBR1 and subsequent pulldowns from the cell lysates using GST-TAX1BP1 immobilized on GSH beads, demonstrated that binding region 2, encompassing the LIR1 of NBR1, is required for the interaction while binding region 1 is not (Fig. 4D). We also confirmed that a small fragment of NBR1 (referred to as CC2-LIR1; aa685– aa757), which contains the binding region 2, alone is sufficient for the interaction with TAX1BP1 in vitro (Fig. EV3B,C). To obtain further insights into the interaction of the NBR1 LIR1 region with the N-domain of TAX1BP1, we performed AlphaFold 2 modeling with sequences of TAX1BP1 and NBR1 encompassing the identified binding sites (Figs. 4E and EV3D)(Jumper et al, 2021). Based on the highest-ranking model, we generated point mutants for both TAX1BP1 and NBR1 and assessed their impact on the interaction of the two proteins. TAX1BP1 point mutants were validated in a microscopy-based interaction assay (Figs. 4F,G and EV3E). NBR1 point mutants were validated by performing a pulldown with cell lysates (Fig. 4H). Residues Y449 and L442 in TAX1BP1 and F740 of NBR1 were identified to be essential for the interaction between the two proteins (Figs. 4F–H and EV3E), consistent with our AF2 model. Given the proximity of residue F740 to the LIR1 motif, we tested whether it is also required for binding to GABARAP/LC3 proteins. To this end, we performed a pulldown with GST-LC3B and detected only a mild reduction in LC3B binding for this mutant (Fig. EV3F).

## GABARAP competes with TAX1BP1 for NBR1 binding

Based on the discovery that the LIR1 region of NBR1 is essential for the binding to TAX1BP1, we speculated that the interaction of

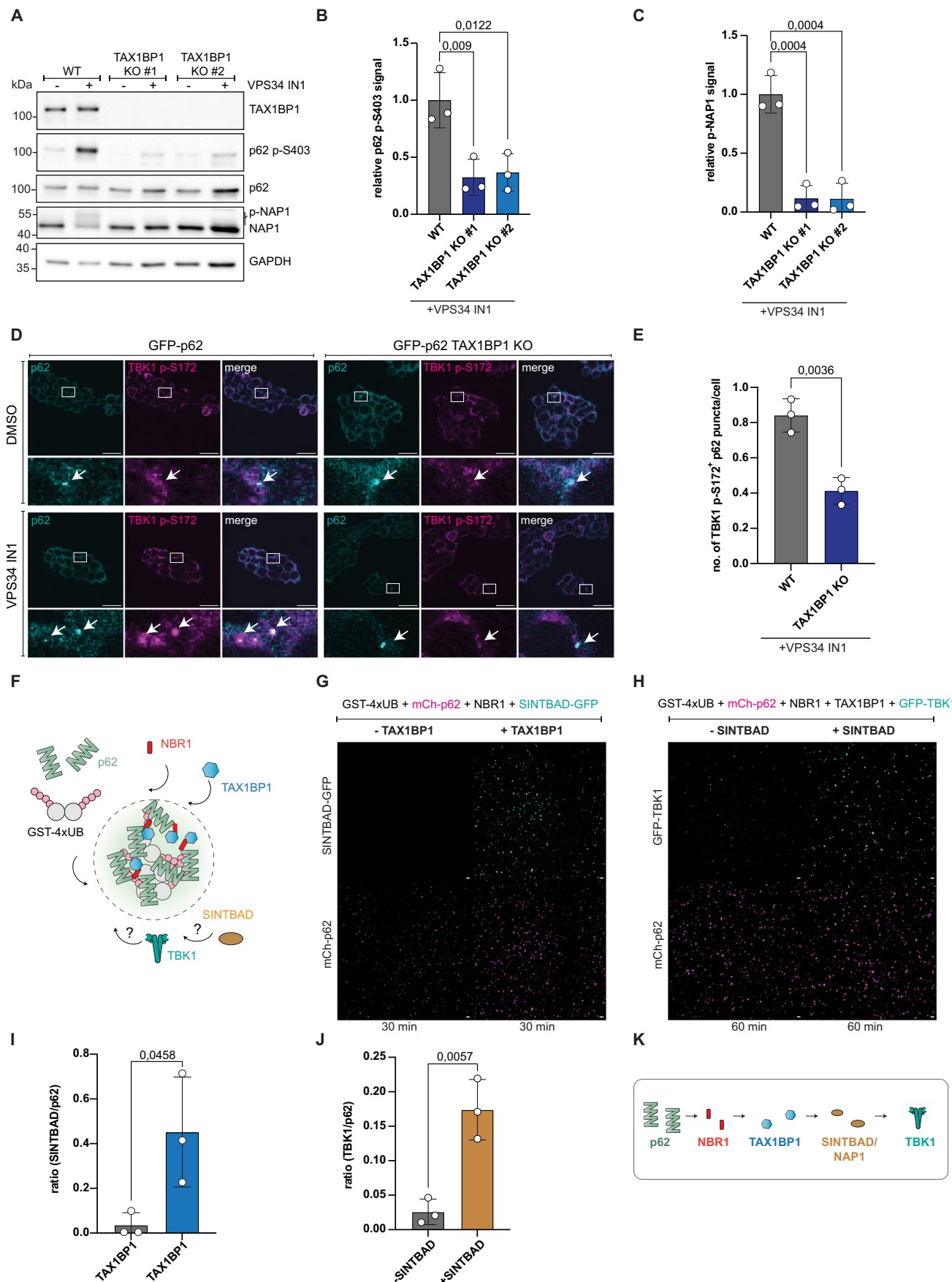

**Figure 3.  TAX1BP1 recruits the TBK1 kinase to p62 condensates via the adapter proteins SINTBAD/NAP1.**

(A) Western blots for p62, phosphorylated p62, and the adapter protein NAP1. (B) Quantification of western blot in (A). Protein levels of phosphorylated p62 upon autophagy inhibition were quantified as a relative to WT. (C) Quantification of western blot in (A). Protein levels of phosphorylated NAP1 upon autophagy inhibition were quantified as a relative to WT. (D) Representative immunofluorescence images of cells stained for p62 and TBK1 Ser172 untreated and upon autophagy inhibition (scale bar = 20 μm). TBK1 p-S172 colocalization with p62 puncta was abrogated in the TAX1BP1 KO cell line (white arrows). (E) Quantification of the number of p62 puncta positive for TBK1 p-S172 upon autophagy inhibition on single focal plane in (D). (F) Scheme of the in vitro condensation assay. (G) Representative image of the in vitro reconstitution of SINTBAD-GFP recruitment to p62 condensates +/− TAX1BP1 (scale bar = 10 μm; loading control in Fig. EV2G). (H) Representative image of the in vitro reconstitution of GFP-TBK1 recruitment to p62 condensates +/− SINTBAD (scale bar = 10 μm; loading control in Fig. EV2H). (I) Quantification of (G). Ratio between number of SINTBAD and p62 condensates is shown at 30 min after the addition of GST-4xUB. (J) Quantification of H. Ratio between number of TBK1 and p62 condensates is shown at 60 min after addition of GST-4xUB. (K) Schematic summary of the results from the condensation experiments. Data in (B, C, E, I, J) show the mean +/− s.d. from three independent experiments. One-way ANOVA with Dunnett's multiple comparison test was performed in (B, C). Unpaired *t* test was performed in (E, I, J). Source data are available online for this figure.

NBR1 with TAX1BP1 and ATG8 proteins such as GABARAP might be mutually exclusive. To test this, we co-incubated GFP-NBR1 coated beads simultaneously with mCherry-TAX1BP1 and increasing amounts of GABARAP. Binding of TAX1BP1 to NBR1 was gradually reduced by the addition of increasing concentrations of GABARAP (Fig. 5A,B; Appendix Fig. S2A). The addition of GABARAP also outcompeted an already established TAX1BP1–NBR1 interaction in vitro (Fig. 5C,D). Next, we asked if GABARAP would also reduce the recruitment of TAX1BP1 to p62 condensates. Indeed, the presence of GABARAP notably reduced the fraction of TAX1BP1-positive p62 condensates in our condensation assay (Fig. 5E,F; Appendix Fig. S2B).

Thus, our findings delineate the interaction between NBR1 and TAX1BP1, revealing that it involves the LIR1 region of NBR1 and that the interaction can be outcompeted by high concentrations of LC3/GABARAP proteins.

## The TAX1BP1–NBR1 interaction is required for the autophagic flux of p62 condensates

To test the relevance of the interaction between TAX1BP1 and NBR1 for the autophagic degradation of p62 condensates in cells, we re-introduced fluorescently tagged wild-type (WT), ΔN (lacking the N-domain) and L442A TAX1BP1 into the TAX1BP1 KO cell line by viral transduction (Fig. EV4A). After doxycycline-induced expression, we observed that in contrast to TAX1BP1 WT (R-TAX1BP1) and consistent with our in vitro data, TAX1BP1 ΔN and L442A (R-TAX1BP1ΔN, R-TAX1BP1 L442A) failed to colocalize with p62 condensates upon treatment with VPS34 IN1 (Fig. 6A,B). Intriguingly, even upon overexpression, TAX1BP1 did not colocalize with p62 puncta and the number of TAX1BP1 foci upon VPS34 IN1 remained lower than for p62 (Figs. 6A and EV4B). Furthermore, re-expression of TAX1BP1 WT rescued the phenotype of the TAX1BP1 KO reducing the number of p62 condensates, while the mutants failed to do so (Fig. 6C). In addition, the TAX1BP1 mutants were unable to restore TBK1-mediated phosphorylation of p62 and the adapter protein NAP1 (Fig. 6D–F). Doxycycline-induced re-expression of TAX1BP1 WT also reduced the NBR1 protein levels, in contrast to the TAX1BP1 mutants (Fig. 6D,G). Moreover, re-expression of TAX1BP1 rescued the recruitment of FIP200 and ATG9 to p62 condensates (Fig. EV4C).

To test the impact of the F740A mutant of NBR1 on TAX1BP1 recruitment, we first generated a GFP-p62 expressing NBR1 KO cell line. Consistent with previous reports, there was a major loss of p62 condensates in the NBR1 KO (Fig. EV4D,E) (Turco et al, 2021). Re-

expression of mScarlet-NBR1 (R-NBR1) rescued the phenotype and enabled p62 condensate formation in the cells (Fig. EV4D,E). Similar to the NBR1-binding deficient TAX1BP1 mutants described above, introduction of the F740A point mutant into NBR1 (R-NBR1 F740A) increased the number of p62 condensates significantly in comparison to the NBR1 WT (Fig. 6H,I). The NBR1 F740A mutant also showed no further increase in the number of p62 condensates upon treatment with the VPS34 inhibitor (Fig. 6H,I). In contrast to NBR1 WT, the point mutant also failed to recruit TAX1BP1 to the p62 condensates (Fig. 6H,J).

Taken together, the interaction between TAX1BP1 and NBR1 is crucial for TBK1 recruitment and to promote autophagy flux of p62 condensates in cells.

## Ubiquitin levels regulate TAX1BP1 recruitment to p62 condensates

Based on the insights we obtained regarding the TAX1BP1–NBR1 interaction, we aimed to understand how the TAX1BP1 recruitment and by implication the initiation autophagosome biogenesis is regulated. p62 condensates harbor two known binding cues for TAX1BP1: ubiquitin, which is bound by TAX1BP1 via its C-terminal ZnF domain (EV5A–C) (Sarraf et al, 2020; Tumbarello et al, 2015) and the region around the LIR1 motif of NBR1, which we identified above. To identify the respective contribution of each individual binding cue for the recruitment of TAX1BP1 to p62 condensates, we reconstituted this step in vitro (Fig. 7A). Deletion of the N-domain completely abolished the recruitment of TAX1BP1 to p62 condensates, while the N-domain alone was still recruited, albeit to a lower level (Figs. 7B,C and EV5D). Conversely, deletion of the ZnF domain resulted in a loss of ubiquitin binding (Fig. EV5A–C) and a decreased TAX1BP1 recruitment to p62 condensates in vitro (Fig. 7B,C), consistent with recent findings (Ferrari et al, 2024). The levels of recruitment of TAX1BP1ΔZnF and of the isolated N-domain were similar, suggesting that the residual recruitment of TAX1BP1ΔZnF is mediated by the N-domain (Figs. 7B,C and EV5D). To further corroborate our in vitro results, we re-introduced TAX1BP1ΔZnF (labeled as R-TAX1BP1ΔZnF) into our TAX1BP1 KO cell line in a doxycycline-inducible manner. In line with our in vitro result, we observed a mild reduction in TAX1BP1 puncta overlapping with p62 and a weaker signal for phosphorylated NAP1 and phosphorylated p62 in comparison to the TAX1BP1 WT (Fig. 7D–H). These results suggest that under the conditions tested, the N-domain-mediated interaction with NBR1 is essential for TAX1BP1 recruitment, while

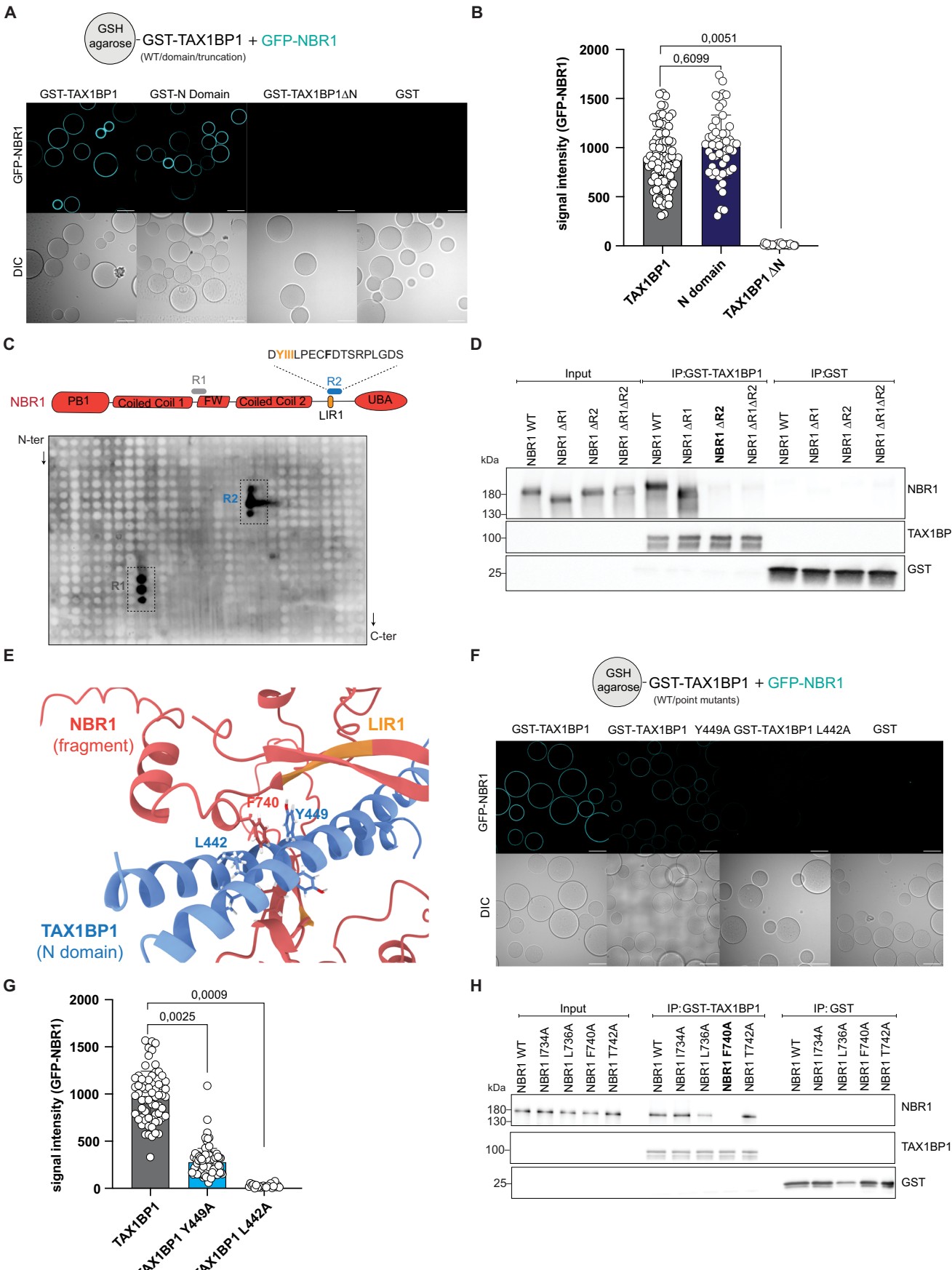

◄ **Figure 4. TAX1BP1 binds to the LIR1 motif of NBR1 via its N-domain.**

(A) Representative images of a microscopy-based protein–protein interaction assay (scale bar = 80 μm; loading control in Fig. EV3A). (B) Quantification of signal intensity of GFP-NBR1 on the beads in (A). (C) Western blot of NBR1 peptide array after incubation with recombinant TAX1BP1 protein. (D) Western blot of a pulldown using lysates derived from HeLa PentaKO cells expressing NBR1 truncations and GST-TAX1BP1 as a bait. (E) AlphaFold 2 model of the interaction between the TAX1BP1 N-domain and the binding region 2 including the LIR1 of NBR1 (pLDDT 44.3 and ptmscore 0.333/iptm 0.387). The region of NBR1 encompassing aa684–aa966 was modeled together with residues aa430–aa468 of TAX1BP1 in a dimeric form using AlphaFold 2. (F) Representative images of a microscopy-based protein–protein interaction assay using TAX1BP1 point mutants (scale bar = 80 μm; loading control in Fig. EV3E). (G) Quantification of the signal intensity of GFP-NBR1 on the beads in (F). (H) Western blot of a pulldown using lysates of HeLa PentaKO cells expressing NBR1 point mutants and GST-TAX1BP1 as a bait. Data in (B, G) show the mean +/− s.d. from three independent experiments. Each data point in (B, G) represent the mean signal intensity for an individual bead. Negative values were excluded and only values above 0 have been plotted and used for the statistical analysis. Nested one-way ANOVA with Dunnett's multiple comparison test was performed in (B, G). Source data are available online for this figure.

the ZnF-mediated recruitment via ubiquitin merely enhances the recruitment or stabilizes the interaction.

Since we found above that the binding of TAX1BP1 to NBR1 is weakened in the presence of GABARAP (Fig. 5), we tested whether the ZnF-mediated interaction with ubiquitin aids in overcoming the competition with GABARAP for efficient TAX1BP1 recruitment. We performed reconstitution experiments using cargo-mimetic beads coated with GST-4xUB and compared the recruitment of GFP-TAX1BP1 full-length and ΔZnF to the beads in the presence of p62, NBR1 and increasing amounts of GABARAP. To ensure that ubiquitin remains a limiting factor and that TAX1BP1 recruitment depends also on NBR1 (as observed in cells (Fig. 6A–C)), we employed low amounts of ubiquitin and pre-coated the beads with mCherry-p62 and NBR1. Indeed, under the conditions tested, the full-length TAX1BP1 protein was robustly recruited to the beads even in the presence of high concentrations of GABARAP while TAX1BP1ΔZnF was barely detectable (Figs. 7I and EV5E). We also performed condensation assays with longer ubiquitin chains (GST-8xUB vs GST-4xUB +/− GABARAP) to mimic a higher ubiquitin load on the cargo and found that more ubiquitin on the cargo rescued TAX1BP1 recruitment to p62 condensates in the presence of GABARAP in vitro (Fig. 7J,K). This suggests that due to their competition with TAX1BP1 for NBR1 binding, the GABARAP/LC3 proteins render the ubiquitin interaction more important for TAX1BP1 recruitment. Hence, we propose that TAX1BP1 is recruited to p62 condensates once a certain local ubiquitin threshold is surpassed, to trigger the switch from cargo collection to autophagosome biogenesis when the cargo load reaches a sufficient level (Fig. 8).

## Discussion

Selective autophagy is a major pathway for the degradation of cellular structures that are out of reach of other degradation systems. For example, it can mediate the degradation of complex organelles such as mitochondria, the endoplasmic reticulum, peroxisomes and even pathogens (Vargas et al, 2023). These cargoes have in common that they represent large structures to which the autophagy machinery can be recruited by cargo receptors to induce autophagosome biogenesis (Ravenhill et al, 2019; Vargas et al, 2019). In this respect aggrephagy is distinct from other selective autophagy pathways, as the cargo material must first be collected into larger condensates before autophagy is triggered. The main collector of the cargo material in aggrephagy is p62 (Bjorkoy et al, 2005; Ciuffa et al, 2015; Kageyama et al, 2021; Sarraf et al,

2020; Sun et al, 2018; Turco et al, 2021; Zaffagnini et al, 2018). Various cellular materials have been shown to be accumulated in the p62 condensates (Zellner et al, 2021) including translation initiation factors (Danieli et al, 2023), aggregation-prone proteins (Bjorkoy et al, 2005), vault complexes (Kurusu et al, 2023) and negative regulators of NRF2 signaling (Kageyama et al, 2021). A prerequisite for the sequestration of most of these materials appears to be their ubiquitination (Bjorkoy et al, 2005; Sun et al, 2018; Zaffagnini et al, 2018). The requirement for cargo collection prior to its sequestration may explain why in aggrephagy at least three cargo receptors cooperate to drive this pathway. Among the three cargo receptors TAX1BP1 is the main recruiter and activator of the autophagy machinery (Sarraf et al, 2020; Turco et al, 2021), although p62 and NBR1 can also bind FIP200 (Turco et al, 2021; Turco et al, 2019). Importantly, for Alzheimer's brain-derived Tau fibrils, the recruitment of TAX1BP1 is compromised (Ferrari et al, 2024).

Given the importance of TAX1BP1 recruitment to cargo material, we set out to understand how its activity can be regulated in cells. We discovered that, in contrast to the in vitro reconstituted p62/NBR1 condensates, TAX1BP1 is not a constitutive component of these structures in cells. This finding suggests that TAX1BP1 recruitment is a major regulatory mechanism. Furthermore, we found that the inhibition of autophagosome biogenesis after its induction using the PI3KC3C1 inhibitor IN1 was required to robustly detect TAX1BP1 at the condensates. Together with the result that deletion of TAX1BP1 results in the accumulation of p62 condensates (Sarraf et al, 2020; Turco et al, 2021), this suggests that its recruitment coincides with the induction of autophagosomes biogenesis. Indeed, in vitro TAX1BP1 is both necessary and sufficient to recruit the TBK1 kinase (Fig. 3F–K), a key factor in selective autophagy as well as FIP200 (Adriaenssens et al, 2024; Matsumoto et al, 2011; Nguyen et al, 2023; Richter et al, 2016; Turco et al, 2021).

We therefore determined the molecular basis for the recruitment of TAX1BP1 to the p62/NBR1 condensates. It was previously shown that a region named N-domain encompassing amino acids 420–506 of TAX1BP1 is required for its interaction with NBR1 (Ohnstad et al, 2020). Furthermore, NBR1 is required for the recruitment of TAX1BP1 to p62 condensates (Turco et al, 2021). We found that the N-domain is indeed sufficient for the binding to NBR1. To our surprise we found that the main binding site for the N-domain of TAX1BP1 in NBR1 is a region encompassing its LIR1 motif, which binds GABARAP/LC3 proteins (Kirkin et al, 2009; Pankiv et al, 2007). The FW domain of NBR1, which we previously showed to weakly bind to TAX1BP1 (Turco et al, 2021) and which also overlaps with binding region 1 (Fig. 4C) did not detectably

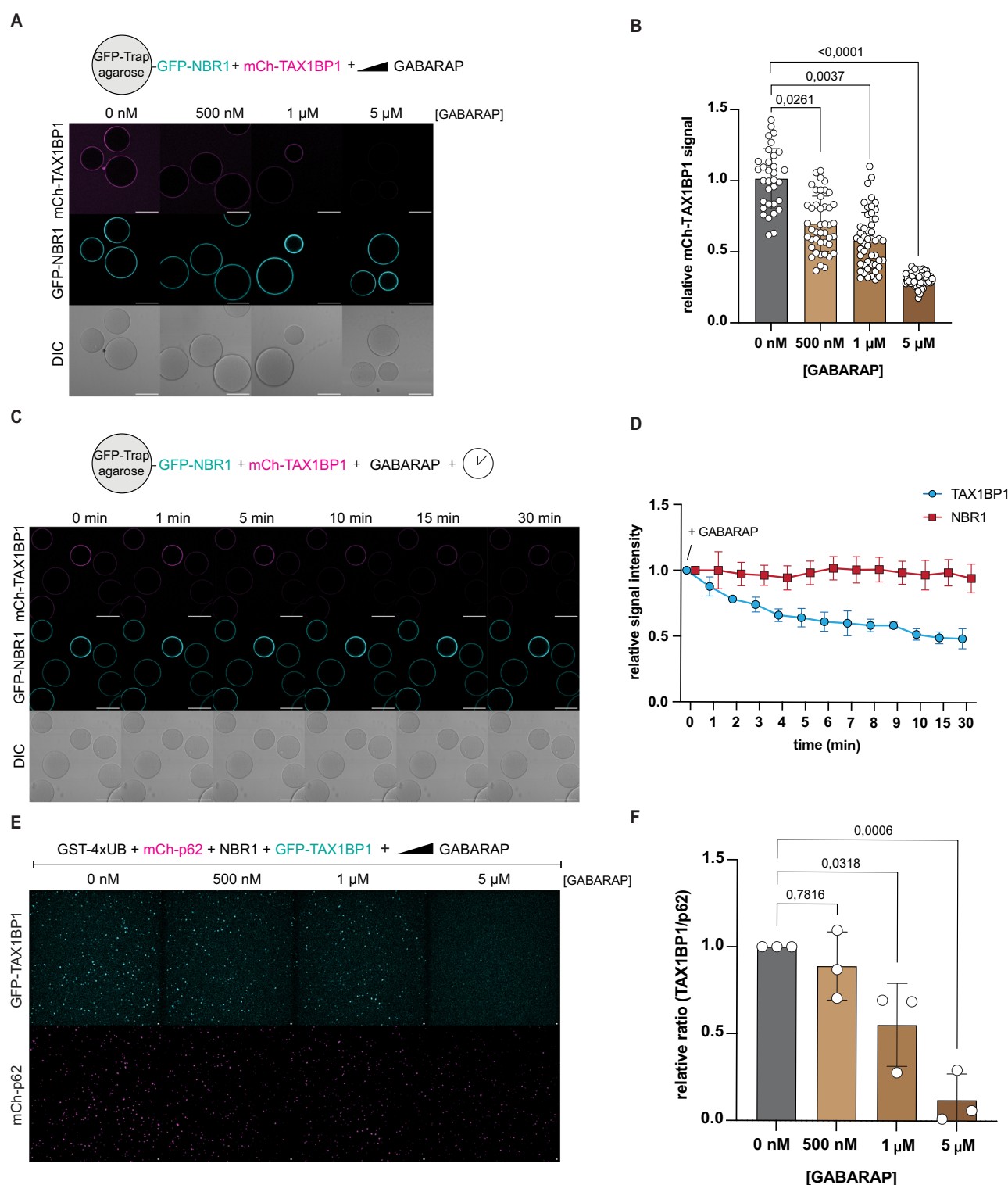

contribute to the interaction with TAX1BP1 in our experiments. The information that the region around the LIR1 of NBR1 is necessary and sufficient for the binding to TAX1BP1 allowed us to model the interaction using AlphaFold 2. While the details of this model will have to be confirmed by structural studies, we could corroborate it through point mutations in both TAX1BP1 and NBR1. Notably, in parallel to us another

study identified the same LIR1 region of NBR1 as the interaction site for TAX1BP1 (North et al, 2024).

The proximity of the TAX1BP1 and GABARAP/LC3 binding sites in NBR1 render the interaction of NBR1 with the two binding partners mutually exclusive (Fig. 5). As a result, the interaction of TAX1BP1 and NBR1 is likely to be weakened in cells, where the

**Figure 5. GABARAP competes with TAX1BP1 for binding to NBR1.**

(A) Representative images of a microscopy-based protein–protein interaction assay adding increasing amounts of GABARAP protein (scale bar = 80 μm; loading control in Appendix Fig. S2A). (B) Quantification of the signal intensity of mCh-TAX1BP1 on the beads in (A) as a relative to the ctrl (0 nM GABARAP). (C) Representative images of a microscopy-based protein–protein interaction assay adding 5 μM GABARAP protein to GFP-NBR1 coupled beads pre-incubated with mCh-TAX1BP1 to monitor the loss of TAX1BP1 signal over time (scale bar = 80 μm). (D) Quantification of mCh-TAX1BP1 signal on the beads in (C) as a relative to timepoint 0 (Addition of GABARAP). The GFP-NBR1 signal was quantified as a bleaching control. (E) Representative images of the in vitro condensation assay adding increasing amounts of GABARAP protein (scale bar = 10 μm; loading control in Appendix Fig. S2B). (F) Quantification of (F). Ratio between number of p62 and TAX1BP1 condensates was plotted as a relative to ctrl (0 nM GABARAP). The data in (B, D, F) show the mean +/− s.d. from three independent experiments. Each data point in (B) represents the mean signal intensity for an individual bead. One-way ANOVA with Dunnett's multiple comparison test was performed in (B, F). Source data are available online for this figure.

concentration of the LC3/GABARAP proteins is considerably higher than that of TAX1BP1 and NBR1 (Cho et al, 2022). This also implies that in cells, TAX1BP1 recruitment might be more dependent on the second binding cue i.e., ubiquitin binding. In fact, NBR1 and ubiquitin binding by TAX1BP1 are both required for full recruitment in vitro as well as in cells and likely also in the brain (Sarraf et al, 2020). We found in a reconstituted system that the presence of GABARAP renders ubiquitin binding by TAX1BP1 essential for its recruitment to cargo-mimetic beads (Fig. 7I–K). We therefore hypothesize that the ubiquitin dependence for its recruitment offers TAX1BP1 the opportunity to sample the ubiquitin load and by implication the cargo load of the p62/NBR1 condensates. Once this load increases above a certain threshold, TAX1BP1 is recruited and further recruits the autophagy machinery including TBK1 via the NAP1 and SINTBAD adapters as well as FIP200 (Turco et al, 2021) to trigger the crucial switch from cargo collection to autophagosome biogenesis (Fig. 8).

It is curious and currently unexplained why the TAX1BP1–NBR1 interaction has evolved to be sensitive to the concentrations of LC3/GABARAP proteins rather than being of lower affinity, which would also allow TAX1BP1 to sense the ubiquitin load. GABARAP/LC3 proteins are themselves consumed by autophagy but also locally concentrated on the nascent autophagosomal membranes. Thus, their overall depletion may globally enhance the NBR1-TAXBP1 interactions while their clustering on the membrane may locally dissociate it, potentially aiding membrane bending and dynamics. Future studies will have to elucidate the molecular gymnastics of these interaction modalities in space and time during aggrephagy.

## Methods

### Reagents and tools table

| Reagent/resource | Reference or source | Identifier or catalog number |
|---|---|---|
| **Experimental models** | | |
| HeLa WT | American Type Culture Collection | RRID: CVCL_0058 |
| HeLa PentaKO | Lazarou et al, 2015 | RRID: CVCL_C2VN |
| HeLa DKO (SINTBAD NAP1 KO) | Adriaenssens et al, 2024 | RRID: CVCL_C9DV |
| HEK293T | American Type Culture Collection | RRID: CVCL_0063 |
| HAP1 | Horizon Discovery | Cat#C631 |

| Reagent/resource | Reference or source | Identifier or catalog number |
|---|---|---|
| HAP1 GFP-p62 | Zaffagnini et al, 2018 | SMcl 117 |
| HAP1 GFP-p62 mSc-NBR1–TAX1BP1-iRFP | This study | SMcl 105 |
| HAP1 GFP-p62 TAX1BP1 KO | This study | SMcl 109 |
| HAP1 GFP-p62 TAX1BP1 KO + TAX1BP1-mSc (alias R-TAXBP1) | This study | SMcl 110 |
| HAP1 GFP-p62 TAX1BP1 KO + TAX1BP1ΔN-mSc (alias R-TAXBP1ΔN) | This study | SMcl 111 |
| HAP1 GFP-p62 TAX1BP1 KO + TAX1BP1 L442A-mSc (alias R-TAXBP1 L442A) | This study | SMcl 112 |
| HAP1 GFP-p62 TAX1BP1 KO + TAX1BP1ΔZnF-mSc (alias R-TAXBP1ΔZnF) | This study | SMcl 113 |
| HAP1 GFP-p62 NBR1 KO | This study | SMcl 114 |
| HAP1 GFP-p62 NBR1 KO + mSc-NBR1 (alias R-NBR1) | This study | SMcl 115 |
| HAP1 GFP-p62 NBR1 KO + mSc-NBR1 F740A (alias R-NBR1 F740A) | This study | SMcl 116 |
| **Recombinant DNA** | | |
| pGEX-GST-4xUB | Gift from Ikeda lab (IMBA) | SMC538 |
| pGEX-GST-8xUB | This study | SMC2505 |
| pET-mCherry-p62 S403E | Zaffagnini et al, 2018 | SMC709 |
| pCAAG-GFP-NBR1 | Ferrari et al, 2024 | SMC1810 |
| pCAAG-NBR1 | Ferrari et al, 2024 | SMC1811 |
| pLIB-10x-His-GFP-TAX1BP1 | Turco et al, 2021 | SMC1435 |
| pLIB-10x-His-GFP-TAX1BP1ΔZnF | This study | SMC2144 |
| pLIB-10x-His-GFP-TAX1BP1ΔN | This study | SMC1569 |
| pLIB-GST-TAX1BP1 | Turco et al, 2021 | SMC1434 |
| pLIB-GST-TAX1BP1ΔZnF | This study | SMC2143 |
| pLIB-GST-TAX1BP1ΔN | This study | SMC1568 |
| pGEX-GST-GFP-TEV-N-domain | This study | SMC2139 |
| pGEX-GST-N-domain | This study | SMC1879 |

| Reagent/resource | Reference or source | Identifier or catalog number |
|---|---|---|
| pLIB-GST-TAX1BP1 Y449A | This study | SMC2244 |
| pLIB-GST-TAX1BP1 L442A | This study | SMC2245 |
| pLIB-10x-His-mCherry-TAX1BP1 | Turco et al, 2021 | SMC1436 |
| pLIB-10x-His-mCherry-TAX1BP1ΔN | This study | SMC1570 |
| pGEX-GST-GFP-CC2-LIR1 | This study | SMC2466 |
| pFBD-GST-SINTBAD | Adriaenssens et al, 2024 | SMC1624 |
| pFBD-GST-SINTBAD-GFP | Adriaenssens et al, 2024 | SMC1613 |
| pFBD-GST-NAP1-mCherry | Adriaenssens et al, 2024 | SMC1628 |
| pFBD-GFP-TBK1 | Adriaenssens et al, 2024 | SMC1631 |
| pGEX-GST-GABARAP | This study | SMC2409 |
| pUC57-TAX1BP1-AID-iRFP-FLAG | This study | SMC1668 |
| AIO-Cas9D10A Puro sgRNA for TAX1BP1 Knock-in | This study | SMC1669 |
| pInducer TAX1BP1-mScarlet | This study | SMC2214 |
| pInducer TAX1BP1ΔN-mScarlet | This study | SMC2216 |
| pInducer TAX1BP1 L442A-mScarlet | This study | SMC2376 |
| pInducer TAX1BP1ΔZnF-mScarlet | This study | SMC2218 |
| AIO Cas9nuc Puro sgRNA for TAX1BP1 Knockout | This study | SMC2302 |
| pCDNA-GFP-NBR1 | This study | SMC2335 |
| pCDNA-GFP-NBR1 ΔR1 (misses entire FW domain) | This study | SMC2338 |
| pCDNA-GFP-NBR1ΔR2 (misses region around LIR1) | This study | SMC2337 |
| pCDNA-GFP-NBR1 ΔR1ΔR2 (misses region around LIR1 + only a part of FW) | This study | SMC2336 |
| pCDNA-GFP-NBR1 I734A | This study | SMC2365 |
| pCDNA-GFP-NBR1 L736A | This study | SMC2366 |
| pCDNA-GFP-NBR1 F740A | This study | SMC2367 |
| pCDNA-GFP-NBR1 T742A | This study | SMC2368 |
| AIO Cas9nuc Puro sgRNA for NBR1 Knockout | This study | SMC2539 |
| pGenLenti-mScarlet-NBR1 | This study | SMC2510 |
| pGenLenti-mScarlet-NBR1 F740A | This study | SMC2511 |
| pGenLenti-mScarlet-mutLIR1 | This study | SMC2512 |
| pGEX-GST-LC3B | Wurzer et al, 2015 | SMC394 |
| pGEX-GST | Addgene: 27458001 | SMC49 |
| VSV-G | Gift from Versteeg lab | SMC1767 |

| Reagent/resource | Reference or source | Identifier or catalog number |
|---|---|---|
| GAG-POL | Gift from Versteeg lab | SMC1768 |
| **Antibodies** | | |
| Anti-p62 | BD Transduction Laboratories | #610832; Clone 3/p62 lck ligand |
| Anti-p62 p-S403 | Cell Signaling Technology | #39786; clone D8D67 |
| Anti-p62 p-S349 | Cell Signaling Technology | #95697 |
| Anti-NBR1 | Abnova | H00004077-M01; Clone 6B11 |
| Anti-TAX1BP1 | Cell Signaling Technology | #5105, clone D1D5 |
| Anti-NAP1 | Sigma | ab192253 |
| Anti-TBK1 p-S172 | Cell Signaling Technology | #5483, clone D52C2 |
| Anti-FIP200 | Cell Signaling Technology | #12436, clone D10D11 |
| Anti-ATG9 | Cell Signaling Technology | #13509, clone D4O9D |
| Anti-GAPDH | Sigma | #G8795, clone 71.1 |
| Anti-GST | Sigma | SAB4200237, clone 2H3-D10 |
| Anti-mouse Alexa 488 | Invitrogen | #A11001 |
| Anti-rabbit Alexa 546 | Invitrogen | #A11010 |
| Anti-rabbit Alexa 647 | Jackson ImmunoResearch | 111-605-144 |
| **Oligonucleotides and other sequence-based reagents** | | |
| NBR1 sgRNA | Lazarou et al, 2015 | CCCAGAGGATCCTGCAGTGC |
| TAX1BP1 sgRNA (for KO) | Lazarou et al, 2015 | AATTGTGTACTAGCATTCCA |
| TAX1BP1 sgRNA 1 (knock-in) | This study | ACACCTAAAATAGACCACTG |
| TAX1BP1 sgRNA 2 (knock-in) | This study | TAGAACATTCTGATCAAAAT |
| siATG14 #1 | GGGGGAGGCAUUAUGUACAAAUA | SMsi 2 |
| siATG14 #2 | GGGGGCUGGAACAAAGAGGUUAAA | SMsi 2 |
| siATG14 #3 | GGGGGCACUUUACAUAGGGUUAA | SMsi 2 |
| siATG14 #4 | GGGGGUAGACUUGAACAGAUAAA | SMsi 2 |
| Primer for pInducer TAX1BP1 constructs | tacccatacgacgtcccagactac | SMP 4959 |
| Primer for pInducer TAX1BP1 constructs | tccaggcgatctgacggttcac | SMP 4960 |
| Primer for pInducer TAX1BP1 constructs | gaaccgtcagatcgcctgga ATGACATCCTTTCAA GAAGTCCCATTG | SMP 4961 |
| Primer for pInducer TAX1BP1 constructs | cccttgctcacTCCGCCacc GTCAAAATTTAGAACATT CTGATCAAAATGGG | SMP 4962 |
| Primer for pInducer TAX1BP1 constructs | CTGGAAAGGTGTACTTGCTCCAC | SMP 4963 |
| Primer for pInducer TAX1BP1 constructs | GTGGAGCAAGTACACCTTTCCAG | SMP 4964 |
| Primer for pInducer TAX1BP1 constructs | gaaccgtcagatcgcctggaATGCGAG CTTCTTCTCCAGTTG | SMP 4965 |
| Primer for pInducer TAX1BP1 constructs | cccttgctcacTCCGCCaccAACATCA AAGCTGGAATCAAAGCAAAAG | SMP 4966 |

| Reagent/resource | Reference or source | Identifier or catalog number |
|---|---|---|
| Primer for pInducer TAX1BP1 constructs | ggtGGCGGAgtgagcaagggcgaggcagtg | SMP 4967 |
| Primer for pInducer TAX1BP1 constructs | tctgggacgtcgtatgggtaTCCGCCaccc ttgtacagctcgtccatgccgc | SMP 4968 |
| Primer for GST-N-domain | ggttccgcgtggatccccggaattcGATCAGG ACAAGACTGATAC | SMP 4116 |
| Primer for GST-N-domain | GGTTCCGCGTGGATCCCCGGAATTCGAT CAGGACAAGACTGATACACTGG | SMP 4117 |
| Primer for pLIB-GST/ GFP-TAX1BP1ΔZnF | CCAGCTTTGATGT TTAGGagctCACTAGTCGCGG | SMP 4428 |
| Primer for pLIB-GST/ GFP-TAX1BP1ΔZnF | CCGTCCCACCATCGGGCG CGATGGTGAGCAAGGGCGAG | SMP 4466 |
| Primer for pLIB-GST/ GFP-TAX1BP1ΔZnF | CCGTCCCACCATCGGGCGCG ATGTCCCTATACTAGGTTATTG | SMP 4467 |
| Primer for pLIB-GST/ GFP-TAX1BP1ΔZnF | CTCGTCGACGTAGGCCTTT GCTAAACATCAAAGCTGGAATC | SMP 4468 |
| Primer for pLIB-GFP-TAX1BP1ΔZnF | CCGTCCCACCATCGGGCG CGGGCCTACGtcgacatgCATCAC | SMP 4477 |
| Primer for TAX1BP1ΔN | CTATGAAAAAACCATCACCTT CTGCAGCAGAGGCAGATTTTG | SMP 3370 |
| Primer for TAX1BP1ΔN | AAGGTGATGGTTTTTTCATAG CATTTAGTTTAAGTTCTGCCACTG | SMP 3371 |
| Primer TAX1BP1 L442A | AAACTCCGTGCTCAGATGGCTG CAGACCATTATAAAGAAAAATTTAAGG | SMP 4852 |
| Primer TAX1BP1 L442A | ATCTGAGCACGGAGTTTCAGAT CTTCAACTTCTCTTCTTAGTTCGAATTC | SMP 4853 |
| Primer TAX1BP1 Y449A | GCAGACCATGCAAAAGAAA AATTTAAGGAATGCCAAAGGCTCC | SMP 4774 |
| Primer TAX1BP1 Y449A | TCTTTTGCATGGTCTGCAGC CATCTGAAGACGGAGTTTC | SMP 4775 |
| Primer for GST-GFP-CC2-LIR1 | GATGGTGATGATGATGGTGA TGcatGAATACTGTTTCCTGTGTGAAATTG | SMP 5409 |
| Primer for GST-GFP-CC2-LIR1 | TCACCATCATCATCACCATC ATCACCACtcaGTGAGCAAGGGCGAGGAG | SMP 5410 |
| Primer for pCDNA-GFP-NBR1 | CTTGGTACCGAGCTCGGATC CgccaccATGgtgagcaagggcgaggagctg | SMP 5145 |
| Primer for pCDNA-GFP-NBR1 | GTGCTGGATATCTGCAGAAT TCtcaatagcgttggctgtaccagtcg | SMP 5146 |
| NBR1ΔR1 | agtctcctttaaaaactgatgatc tcacctgccagcaagag | SMP 5147 |
| NBR1 F740A | tgagtgcGCTgatacc agccgcccctgg | SMP 5198 |
| NBR1 F740A | ggtatcAGCgcactcag gcaggatgatgatgtaatcctc | SMP 5199 |
| **Chemicals, enzymes, and other reagents** | | |
| **Software** | | |
| Image J2 version 2.14.0 | | |
| GraphPad Prism 10 | | |
| **Other** | | |

10% (v/v) fetal bovine serum (FBS, Thermo Fisher), 25 mM HEPES (15630080, Thermo Fisher), 1% (v/v) non-essential amino (NEAA, 11140050, Thermo Fisher), and 1% (v/v) penicillin–streptomycin (15140122, Thermo Fisher). HAP1 cells were cultured in Iscove's Modified Dulbecco's Medium (IMDM, 31980-022, Thermo Fisher), supplemented with 10% (v/v) fetal bovine serum (FBS, Thermo Fisher) and 1% (v/v) penicillin–streptomycin (15140122, Thermo Fisher). Cell lines were regularly tested for mycoplasma contaminations.

Sf9 insect cells (12659017, Thermo Fisher) were cultured as a shaking liquid culture at 27 °C in insect culture medium (96-001-01, ESF 921 Expression System LLC) supplemented with 1% penicillin–streptomycin (15140122, Thermo Fisher).

## Reagents and cell treatments

The following chemical compounds have been used in this study: Bafilomycin A1 at 400 nM for 3 h (sc-201550, Santa Cruz Biotech), Doxycycline at 50 ng/ml for 16 h (D9891, Sigma), DMSO (D2438, Sigma), Puromycin at 5 μg/ml (A11138-03, Thermo Fisher), VPS34 IN1 at 2 μM for 3 h (APE-B6179, ApexBio). siRNAs were purchased from Dharmacon as a pool of 4 individual siRNAs and used at a final concentration of 20 nM.

## Cloning

The sequences for all the inserts were generated by amplifying existing plasmids or through gene synthesis (GenScript). Plasmids were generated either by Gibson assembly or gene synthesis. For Gibson assembly, the vector and inserts were generated by restriction digestion or PCR. DNA was cleaned up with Promega Wizard SV gel and PCR Cleanup System (A2891, Promega). The inserts and the backbone were mixed in a 3:1 ratio and incubated with the 2x NEBuilder HiFi DNA assembly enzyme mix (M5520A, New England Biolabs) for 1 h at 50 °C. Competent DH5a *E.coli* were transformed with the Gibson reaction and incubated overnight at 37 °C. The next day single colonies were picked, grown overnight, and plasmids were isolated using the GeneJet Plasmid Miniprep Kit (K0503, Thermo Fisher). For sequence verification, the isolated plasmids were sent for Sanger sequencing (MicroSynth AG).

## CRISPR/Cas9 genome editing

sgRNAs for generating HAP1 GFP-p62 mSc-NBR1–TAX1BP1-iRFP cell line were designed by CRISPOR (http://crispor.tefor.net). Both sgRNAs were cloned into an all-in-one vector encoding Cas9D10A nickase (Addgene) with puromycin as a selection marker. The plasmid bearing the repair template was generated by gene synthesis (GenScript). All-in-one vector and repair template were transfected into GFP-p62 mSc-NBR1 (previously validated in (Turco et al, 2021)) using Fugene 6 (E2691, Promega). After selection with puromycin, the remaining cells were single-cell sorted by FACS and validated by PCR, western blot and Sanger sequencing. The selected clone was also validated for protein levels and autophagy flux.

The same sgRNAs for generating the TAX1BP1 KO and NBR1 KO cell line has been used as previously reported (Lazarou et al, 2015; Sarraf et al, 2020) and cloned into pSpCas9(BB)-2A-Puro

## Cell culture

All mammalian cell lines were cultured at 37 °C in a humidified 5% $CO_2$ atmosphere. HeLa PentaKO cells were obtained from Michael Lazarou (Lazarou et al, 2015). HEK293T cells were purchased from the American Type Culture Collection (ATCC). HAP1 cells were originally purchased from Horizon Discovery. GFP-p62 and GFP-p62 mSc-NBR1 were previously generated and used as parental cell lines in this study (Turco et al, 2021; Zaffagnini et al, 2018). HeLa and HEK293T cells were grown in Dulbecco Modified Eagle Medium (DMEM, 41966-029, Thermo Fisher) supplemented with

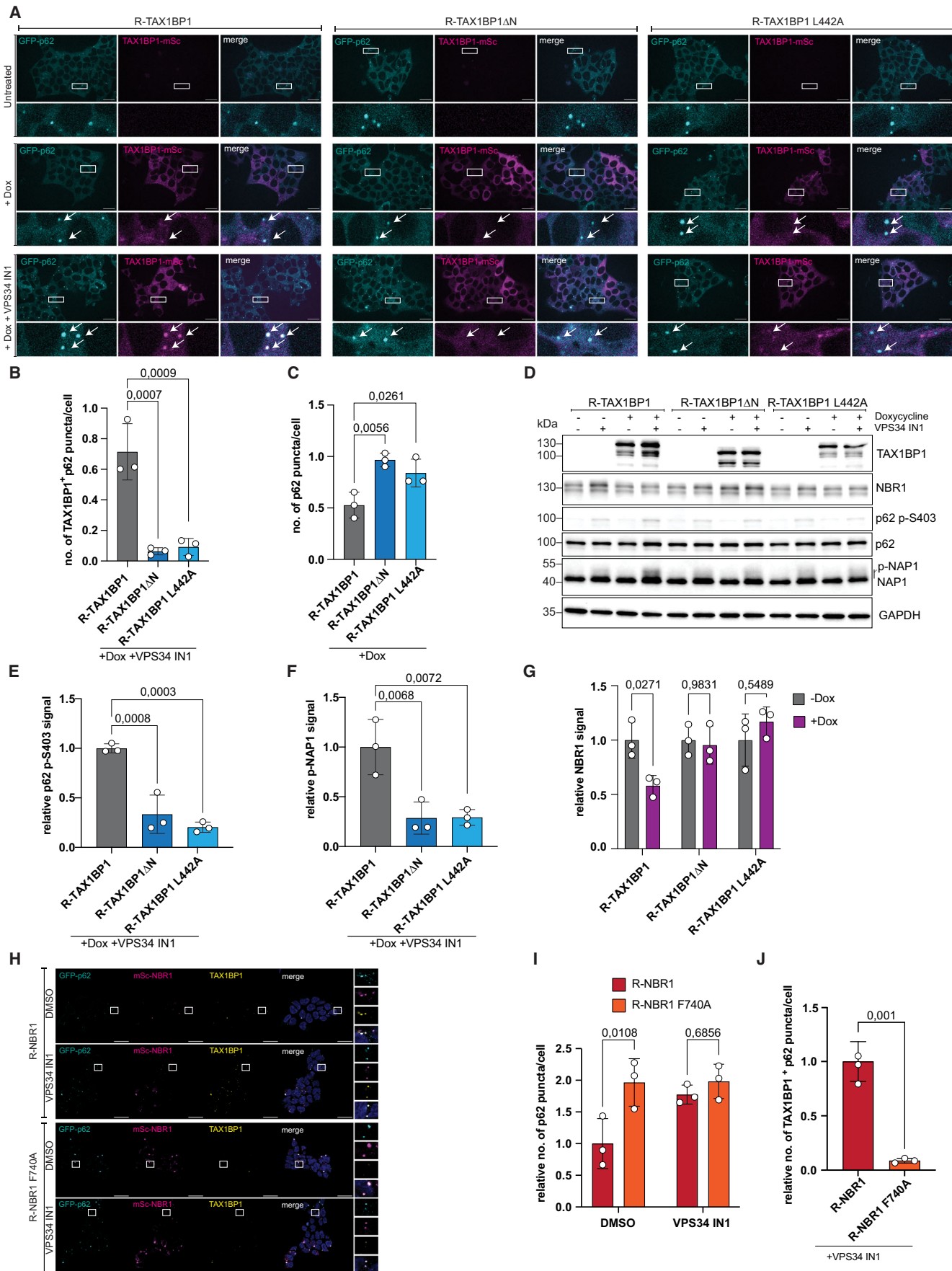

◄ **Figure 6.  The TAX1BP1–NBR1 interaction is required for the autophagic flux of p62 condensates in cells.**

(A) Representative live-cell images of TAX1BP1 KO cells expressing endogenously tagged GFP-p62 either untreated, treated with doxycycline or treated with doxycycline and VPS34 IN1 (scale bar = 20 µm). TAX1BP1 WT was recruited to p62 puncta upon the addition of doxycycline and VPS34 IN1 in contrast to TAX1BP1ΔN and TAX1BP1 L442A (white arrows). (B) Quantification of the number of p62 puncta positive for TAX1BP1 upon addition of doxycycline and VPS34 IN1 on a single focal plane from (A). (C) Quantification of the total number of p62 puncta upon the addition of doxycycline on a single focal plane from (A). (D) Western blot of cells expressing TAX1BP1 WT, ΔN or L442A upon doxycycline treatment and autophagy inhibition. (E) Quantification of the western blot in (D). The phosphorylation levels of p62 upon VPS34 IN1 treatment were quantified as a relative to R-TAX1BP1. (F) Quantification of western blot in (D). The phosphorylation levels of NAP1 upon autophagy inhibition were quantified as a relative to R-TAX1BP1. (G) Quantification of western blot in (D). The NBR1 protein levels were quantified as a relative to the corresponding -doxycycline samples. (H) Representative immunofluorescence images of NBR1 KO cells re-expressing mSc-NBR1 WT or mSc-NBR1 F740A (scale bar = 20 µm). (I) Quantification of the number of p62 puncta in (H) per cell and focal plane. The number of p62 puncta was plotted as a relative to R-NBR1 DMSO treated samples. (J) Quantification of number of TAX1BP1-positive p62 puncta in (H) per cell and focal plane. The number of TAX1BP1-positive p62 puncta was plotted as a relative to R-NBR1. The data in (B, C, E, F, G, I, J) show the mean $+/-$ s.d. from three independent experiments. One-way ANOVA with Dunnett's multiple comparison test was performed in (B, C, E, F). Two-way ANOVA with Sidak's multiple comparison test was performed in (G, I). Unpaired $t$ test was perfomed in (J). Source data are available online for this figure.

vector backbone. The plasmids were transfected into HAP1 GFP-p62 cells (previously described in (Zaffagnini et al, 2018)). After selection with puromycin, cells were single-cell sorted by FACS. Individual clones were validated using PCR, western blots and Sanger sequencing.

## Generation of stable cell line

For stable cell line generation, lentiviral transductions were performed. HEK293T cells were transfected with VSV-G, GAG-POL and lentiviral plasmids carrying the transgene using Lipofectamine 3000 (L3000008, Thermo Fisher). After 24 h, the medium of the transfected cells was exchanged to the appropriate medium for the target cells. HAP1 GFP-p62 TAX1BP1 KO and HAP1 GFP-p62 NBR1 KO cells were seeded on a 6 well plate at 400k cells/well. 24 h later the medium containing the lentivirus was collected and filtered with a 0.45-µm filter (729025, Macherey-Nagel). In total, 8 mg/ml polybrene (Sigma) was added to the viral supernatant. The target cells were incubated for 24 h with the virus containing medium. The cells were washed with PBS on the next day and virus-free medium was added back to the cells. After selection with antibiotics, the expression of the transgene was validated by western blot.

## Western blots

For western blot analysis, cells were harvested using trypsin and washed 2x with PBS. Cells were resuspended in an appropriate volume of lysis buffer (50 mM Tris-HCl pH 7.4, 1 mM EGTA, 1 mM EDTA, 1% Triton X-100, 0.27 M sucrose, 1 mM DTT, cOmplete EDTA-free protease inhibitor (11836170001, Roche)) and incubated for 30 min on ice. Lysates were cleared by centrifugation and protein concentration was determined by Bradford (Biorad). Overall, 10 µg of protein was boiled for 5 min at 95 °C and then loaded on a 4–15% SDS gel (4561086, Biorad).

Proteins were transferred on a nitrocellulose membrane (10600001, Cytiva) by wet blotting (Biorad) at 120 V for 60 min. Membranes were blocked with 3% non-fat dry milk in TBS + 0.1% Tween-20 (blocking buffer) for 1 h at RT. Primary antibodies were added for overnight incubation at an appropriate dilution (see below). The next day membranes were washed 3× with TBS + 0.1% Tween-20 (TBST) and incubated with the secondary antibody diluted in blocking buffer. After three washes with TBST, membranes were developed with Super Signal West femto chemiluminescence substrate (34096, Thermo Fisher). Images were

taken with a ChemiDoc MP Imaging system (Bio-Rad). Antibodies dilutions used for western blot: rabbit anti-AZI2/NAP1 (Abcam 1:1000), mouse anti-NBR1 (Abnova 1:1000), mouse anti-GAPDH (Sigma, 1:25,000), mouse anti-p62 (BD Bioscience, 1:1000), rabbit anti-p62 p-S349 (Cell Signaling, 1:1000), rabbit anti-p62 p-S403 (Cell Signaling, 1:1000), rabbit anti-TAX1BP1 (Cell Signaling, 1:1000). Secondary antibodies: goat anti-rabbit HRP (Jackson Immunoresearch, 1:10,000), goat anti-mouse HRP (Jackson Immunoresearch, 1:10,000).

For quantification, ImageJ was used. A rectangle was drawn around the corresponding protein band and the intensity was plotted using the "Plot Lanes" function. Measured values were adjusted and corrected by GAPDH as a loading control.

## Live-cell imaging

Cells were seeded in 10-well glass-bottom imaging chambers (543079, Greiner Bio-One) at 6000 cells/ well. 48 h later, cells were treated and imaged using a Visitron Live Spinning Disk microscope (Plan-Apochromat 63×/1.4 Oil DIC objective and an EM-CCD camera; including a temperature & CO₂-controlled environment). Standard imaging settings were a gain of 500 for all channels and 100 ms exposure for GFP & mScarlet and 200 ms for iRFP. Laser settings were the following: 5% laser power for GFP, 10% laser power for mScarlet and 10% laser power for iRFP. For dox-inducible TAX1BP1 constructs, iRFP settings were a gain of 500, 150 ms exposure and 15% laser power. Settings were kept consistent across replicates. Multiple areas within the wells were imaged to obtain comparable numbers of cells (50–150 cells). No blinding was performed during image acquisition.

For quantifications a macro for ImageJ was used as previously described (Turco et al, 2021). In short, binary pictures were generated by thresholding. Analyze punctae with a cut off of ">3 pixel units" was used to count puncta and to save coordinates as ROIs. ROIs from one channel were overlayed with the other channel(s) to determine the degree of colocalization. Threshold was determined manually and was kept consistent across replicates. Cells with a significant amount of background noise were excluded from the analysis.

## Immunofluorescence and confocal microscopy

Immunofluorescence was performed similar to previous studies (Turco et al, 2021). In short, cells were seeded in six-well plates on coverslips (0117520, Marienfeld Superior). After 24 h the cells were

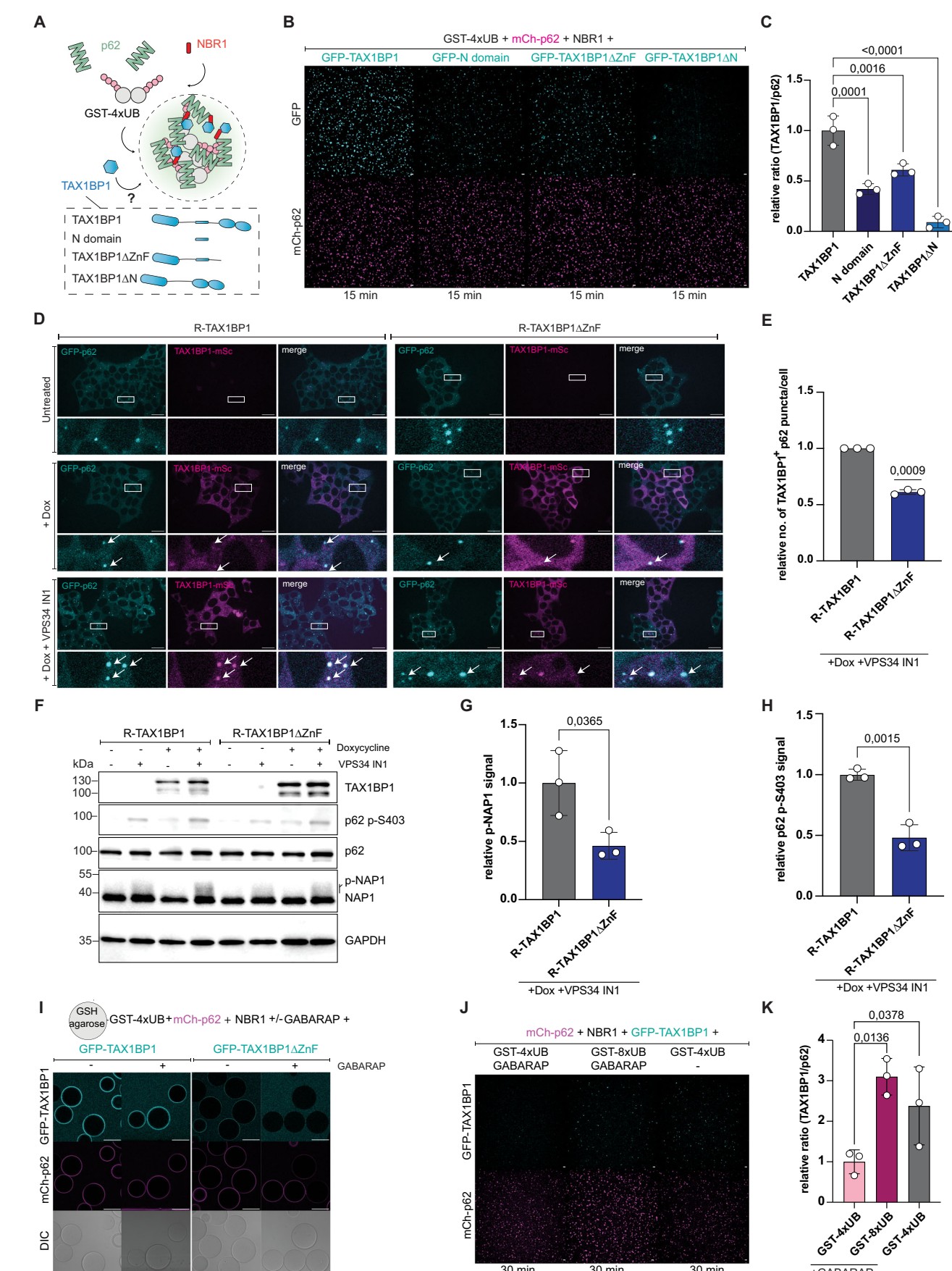

Figure 7.   Ubiquitin levels regulate TAX1BP1 recruitment to p62 condensates.

(A) Scheme of the in vitro condensation assay comparing the recruitment of GFP-TAX1BP1 truncations or domains to p62 condensates. (B) Representative images of the in vitro condensation assay comparing the recruitment of TAX1BP1 full-length (FL) vs ΔN vs ΔZnF (scale bar = 10 μm; loading control in Fig. EV5D). (C) Quantification of (B). The ratio of the number of p62 condensates and GFP-TAX1BP1 (FL/N-domain/TAX1BP1ΔZnF/TAX1BP1ΔN) was plotted as a relative to GFP-TAX1BP1. (D) Representative live-cell images of TAX1BP1 KO cells expressing endogenously tagged GFP-p62 either untreated, treated with doxycycline or treated with doxycycline and VPS34 IN1 (scale bar = 20 μm). TAX1BP1 ΔZnF was recruited less to p62 puncta upon the addition of doxycycline and VPS34 IN1 in comparison to TAX1BP1 WT (white arrows). The same image for the WT control has been used as in Fig. 6A. (E) Quantification of the number of p62 puncta positive for TAX1BP1 upon addition of doxycycline and VPS34 IN1 as a relative to R-TAX1BP per cell and on a single focal plane from (D). (F) Western blot of cells expressing TAX1BP1 WT or ΔZnF upon doxycycline treatment and autophagy inhibition with VPS34 IN1. (G) Quantification of the western blot in F. The NAP1 phosphorylation levels upon VPS34 IN1 treatment were quantified as a relative to R-TAX1BP1. (H) Quantification of western blot in (F). The p62 phosphorylation levels upon VPS34 IN1 treatment were quantified as a relative to R-TAX1BP1. (I) Representative images of microscopy-based interaction assay to compare GFP-TAX1BP1 (WT and ΔZnF) recruitment to cargo-mimetic beads in the presence of GABARAP (scale bar = 80 μm; loading control in Fig. EV5E). (J) Representative images of the in vitro condensation assay comparing the recruitment of GFP-TAX1BP1 in the presence of either GST-4xUB or GST-8xUB with and without GABARAP (scale bar = 10 μm). (K) Quantification of condensation assay in (J). The normalized ratio of TAX1BP1-positive to p62 condensates was plotted as a relative to GST-4xUB + GABARAP. The data in (C, E, G, H, K) show the mean +/− s.d. from three independent experiments. One-way ANOVA with Dunnett's multiple comparison test was performed in (C). One-sample t test and Wilcoxon test was performed in (E). Unpaired t test was performed in (G, H). One-way ANOVA with Holm–Sidak's multiple comparison test was performed in (K). Source data are available online for this figure.

treated, washed 3× with PBS and fixed with 4% Paraformaldehyde in PBS for 20 min at RT. After fixation, the cells were washed 3× with PBS and permeabilized with 0.1% Triton X-100 in PBS for 5 min at RT except for experiments that included FIP200. For FIP200, cells were permeabilized with 0.25% Triton X-100 in PBS for 15 min at RT. Cells were washed 2× with PBS, incubated with blocking buffer (1% BSA in PBS) at RT for 1 h and incubated with the primary antibodies for 1 h at RT in a humid chamber except for FIP200. For FIP200, coverslips were incubated immediately with the primary antibody in a humid chamber for 1 h at 37 °C without a previous blocking step. The cells were washed 3x with PBS and incubated with the secondary antibody for 1 h at RT in a humid chamber. Cells stained with FIP200 were incubated for 1 h at 37 °C in a humid chamber with the secondary antibody. All cells were washed 3× with PBS and mounted on glass slides (AA00000112E01MNZ10, epredia) by transferring them onto a droplet of the mounting medium DAPI-Fluoromont-GTM (0100-200, Southern Biotech).

Antibody dilutions (in blocking buffer): rabbit anti-ATG9 (1:100, Cell Signaling), rabbit anti-FIP200 (1:200, Cell Signaling), mouse anti-p62 (1:100, BD Bioscience), rabbit anti-AZI2/NAP1 (1:100, Abcam), rabbit anti-TAX1BP1 (1:100, Cell Signaling), anti-TBK1 p-S172 (1:100, Cell Signaling), goat anti-rabbit Alexa Fluor 546 (1:1000, Invitrogen), goat anti-mouse Alexa Fluor 488 (1:1000, Invitrogen), goat anti-rabbit Alexa Fluor 647(1:500, Jackson ImmunoResearch).

Images were taken with a Zeiss LSM 700 confocal microscope equipped with Plan-Apochromat 63×/1.4 Oil DIC objective. Multiple areas within samples were imaged to obtain comparable numbers of cells (50–150 cells). No blinding was performed during image acquisition. To avoid cross-contamination between fluorochromes, each channel was imaged sequentially using the multitrack recording module before merging. Images from fluorescence and confocal acquisitions were processed and analyzed with ImageJ software.

For quantification, a macro for ImageJ was used as described before (Turco et al, 2021). In short, binary pictures were generated by thresholding. Analyze puncta with a cut off of ">3 pixel units" was used to count the number of puncta and to save coordinates as ROIs. ROIs from one channel were overlayed with the other channel(s) to determine the degree of colocalization. The threshold was determined manually and was kept consistent across replicates.

Cells with a significant amount of background noise were excluded from the analysis.

## Protein production

N-domain (aa420-506 of TAX1BP1), GFP-N-domain, Tetra-ubiquitin (4xUB), GFP and GFP-CC2-LIR1 were cloned into a pGEX -4T1 plasmid vector backbone. mCherry-p62 S403E was cloned into the pET vector backbone. Octa-ubiquitin (8xUB) was generated by gene synthesis and cloned into the pGEX-4T1 vector backbone by GenScript. After transformation into competent E. coli Rosetta pLysS cells (70956-4, Novagen), a single colony was picked and incubated overnight in 100 ml of LB including the appropriate selection marker. The next day, the cultures were scaled up to the range of liters. The cultures were incubated at 37 °C until an $OD_{600}$ of 0.6 was reached and then the expression was induced by addition of 0.2 mM isopropylthiogalactoside (IPTG). The cells were harvested after 18–20 h at 18 °C and pellets were flash-frozen in liquid nitrogen and stored at −70 °C. For mCherry-p62 S403E, the cells were incubated until they reached the OD 0.8. The expression was induced with 0.2 mM IPTG and the cells were harvested after 5 h shaking at 25 °C, flash-frozen and stored at −70 °C.

10x-His-GFP-TAX1BP1, 10x-His-GFP-TAX1BP1ΔN, 10x-His-GFP-TAX1BP1ΔZnF, 10x-His-mCherry-TAX1BP1, 10x-His-mCherry-TAX-1BP1ΔN, GST-TAX1BP1, GST-TAX1BP1ΔZnF, GST-TAX1BP1ΔN, GST-TAX1BP1 L442A, GST-TAX1BP1 Y449A were cloned into the pLIB vector backbone for insect cell culture expression. Codon-optimized GFP-TBK1, TBK1, SINTBAD and SINTBAD-GFP were generated by gene synthesis (GenScript) for insect cell expression. The plasmids were transformed into electrocompetent DH10BacY cells (Vijayachandran et al, 2011). For the transformation, 1 μl of the plasmids at a concentration of 25 ng/μl were mixed with bacteria and electroporated using a Biorad Micropulser at the appropriate settings for bacteria (2.5 kV potential difference). For recovery, the electroporated cells were mixed with 5 ml of LB and incubated for at least 5 h at 37 °C. Afterward, the cells were plated on LB agar plates containing 50 μg/ml Kanamycin, 7 μg/ml Gentamicin, 10 μg/ml Tetracycline, 200 μg/ml X-Gal and 165 μM IPTG. White colonies were screened on the next day by colony PCRs. Positive colonies were picked and incubated with 10 ml of LB and appropriate antibiotics overnight at 37 °C. The next day the bacmids were isolated and the DNA measured. The bacmids were used freshly for insect cell transfections. 1 million of Sf9 insect cells were seeded on 6 well plates. 5 μg of DNA and 5 μl

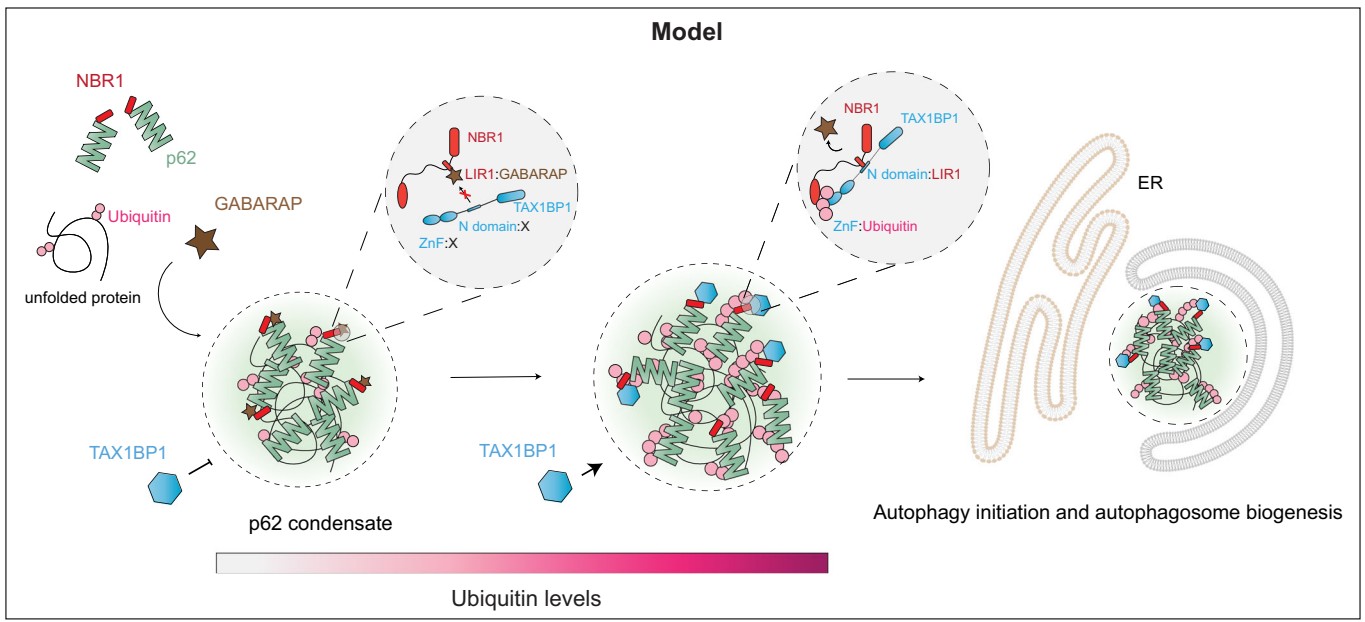

**Figure 8.** Model for the ubiquitin load dependent, TAX1BP1-mediated switch from cargo collection to autophagosome biogenesis in aggrephagy.

of FuGene HD (E2311, Promega) was used for transfection. The expression of the reporter YFP was tracked over several days by microscopy. Once most cells showed YFP expression (usually after 5 days), the cell suspension (V0) was added to 30 ml of insect cells at 1 mio/ml. Viability of cells and YFP expression was monitored for several days. Once, the cells showed viability drop to below 90%, the cells were spun down and the supernatant was filtered with a 0.45-μm filter (729025, Macherey-Nagel) and stored at 4 °C. For expression, 1 ml of the V1 was added to 900 ml of Sf9 insect cells at 1 mio/ml. Once the viability dropped below 90%, the cells were harvested, washed once with PBS, flash-frozen and stored at −70 °C.

GFP-NBR1 and NBR1 were cloned into the pCAGGSsH-C vector and expressed in HEK293 via the VBCF ProTech facility (https://www.viennabiocenter.org/vbcf/protein-technologies/). Pellets were stored at −70 °C.

## Protein purification

mCherry-p62 S403E, GST-4xUB, GFP-NBR1, NBR1, GFP-TAX1BP1 FL, mCherry-TAX1BP1 FL were purified as described previously (Ferrari et al, 2024; Turco et al, 2021; Zaffagnini et al, 2018).

Truncations and point mutants of TAX1BP1 were purified following the same procedure as for the full-length TAX1BP1 (FL) protein.

GFP-TBK1, TBK1, SINTBAD, and SINTBAD-GFP were also purified as previously described (Adriaenssens et al, 2024).

GST-N-domain, GFP-N-domain, GST-GFP-CC2-LIR1, and GST-8xUB were purified from bacterial pellets. Pellets corresponding to 2 L culture were resuspended in lysis buffer (50 mM HEPES, pH 7.5, 300 mM NaCl, 1 mM DTT, 2 mM MgCl$_2$, cOmplete EDTA-free protease inhibitor (Roche), DNase (Sigma),) and lysed by sonication. To clear the lysate, the suspension was centrifugated with 18,000 rpm at 4 °C. The supernatant was filtered through a

0.45-μm filter (6880-2504, Cytiva) and incubated with 3 ml pre-equilibrated GSH beads (17075605, GE Healthcare). After 1 h incubation rolling at 4 °C, the beads were washed 5× with Wash Buffer I (50 mM HEPES, pH 7.5, 300 mM NaCl, 1 mM DTT), once with Wash Buffer II (50 mM HEPES, pH 7.5, 700 mM NaCl, 1 mM DTT) and again 2× with Wash Buffer I. For elution, beads were either resuspended in 3 ml Wash Buffer I + TEV protease for cleavage overnight or by incubation with elution buffer (Wash Buffer I + 50 mM glutathione (Sigma) - pH adjusted to 7.5). The next day, the eluate was filtered through a 0.45-μm filter, concentrated to 500 μl using an appropriate protein concentrator (Merck Millipore), and further purified by size exclusion chromatography on a Superdex 200 Increase 10/300 column (Cytiva) in SEC Buffer (25 mM HEPES pH 7.5, 150 mM NaCl, 1 mM DTT).

## Microscopy-based protein interaction assay

Microscopy-based interaction assay was performed as previously described (Sawa-Makarska et al, 2020; Turco et al, 2021; Zaffagnini et al, 2018). In short, GSH (17075601, GE Healthcare), GFP-trap or RFP-trap beads (gta & rta, Chromotek) were washed first 2x with H$_2$O and then equilibrated 2× with SEC buffer (25 mM HEPES, pH 7.5, 150 mM NaCl, 1 mM DTT). Beads were resuspended in 40 μl of SEC buffer, and the bait protein was added. For experiments with GST-TAX1BP1 (FL, domain, truncations, point mutants) in Fig. 4A, 2.5 μM of bait protein was added to the beads. The beads were incubated for 1 h at 4 °C and washed afterwards 3× with SEC Buffer. The beads were pipetted into a 384-well glass-bottom plate (781892, Greiner Bio-One) containing 1 μM GFP-NBR1 as prey in a total volume of 20 μl. For experiments in Fig. EV5A, 5 μM of GST-4xUB as bait protein was added and treated as described above. Beads were pipetted into a 1 μM GFP-TAX1BP1 (FL, truncation) containing prey solution. For experiments in Fig. 5A,C, GFP-trap beads were incubated with 2.5 μM GFP-NBR1 and

treated as described above. Beads were added to 500 nM mCherry-TAX1BP1 and increasing amounts of GABARAP protein (0 nM–500 nM–1 µM–5 µM). For experiments in Fig. 7I, GSH beads were first incubated with a mix of 4.5 µM GST + 0.5 µM GST-4xUB as a bait for 1 h. After the incubation, the beads were washed to remove the unbound proteins and further incubated with 1 µM mCh-p62 and 1 µM NBR. After 30 min of incubation, 500 nM GFP-TAX1BP1 FL or GFP-TAX1BP1ΔZnF, together with 5 µM GABARAP, was added to the beads. Prey & bait were incubated for 15 min to 30 min at RT and imaged with a Zeiss LSM 700 confocal microscope equipped with Plan-Apochromat 20X/0.8 WD 0.55 mm objective. For each experimental condition, three biological replicates were performed. For the quantification, we employed an artificial intelligence (AI) script that automatically quantifies signal intensities from microscopy images by drawing line profiles across beads and recording the difference between the minimum and maximum gray values along the lines. The AI was trained to recognize beads employing cellpose (Stringer et al, 2021). Source code available at https://www.maxperutzlabs.ac.at/research/facilities/biooptics-light-microscopy. Background-corrected values were plotted and subjected for statistical analysis. For the GABARAP competition assay, we normalized to the control (0 nM GABARAP) to minimize the inter-experiment variability.

## Pulldowns

For pulldowns with cell lysates, HeLa PentaKO cells were seeded at 500k/well. Twenty-four hours later, the cells were transfected with plasmids encoding GFP-NBR1 (FL, truncations, point mutants). Forty-eight hours after transfections, cells were harvested and lysed using lysis buffer (50 mM Tris-HCl pH 7.4, 1 mM EGTA, 1 mM EDTA, 1% Triton X-100, 0.27 M sucrose, 1 mM DTT, cOmplete EDTA-free protease inhibitor (Roche)). GST-TAX1BP1 coated beads were prepared as described above. In total, 100–200 µg of total protein lysates was mixed with beads and incubated for 1 h at 4 °C. After the incubation, beads were washed 3× with IP wash buffer (20 mM Tris-HCl pH 7.4, 150 mM NaCl, 1 mM EDTA, 0.05% Triton X-100, 5% glycerol) to remove unbound material. Overall, 40 µl of 2× loading dye was added to the beads. Samples were boiled for 10 min at 95 °C. SDS PAGE and western blotting for detection were performed as described above.

## Condensate assay

In vitro condensation assays have been performed as previously described (Turco et al, 2021; Turco et al, 2019; Zaffagnini et al, 2018). In short, proteins were spun down with max speed for 10 min at 4 °C. For experiments in Figs. 1E and 7B, 2 µM mCherry-p62 S403E, 1 µM NBR1, 2 µM GFP-TAX1BP1 (FL, ΔN or ΔZnF) were mixed in SEC buffer (25 mM HEPES, pH 7.5, 150 mM NaCl, 1 mM DTT) in a 384-well glass-bottom plate (781892, Greiner Bio-One). Buffer without NaCl was used to keep the salt level at 150 mM NaCl. 5 µM GST-4xUB was added at the microscope to trigger condensation of the proteins. Images were taken as a time course over 2 h with a Spinning disk microscope equipped with LD Achroplan 20X/0.4 Corr objective and EM-CCD camera.

For the reconstitution of the recruitment of TBK1 & SINTBAD to p62 condensates in Fig. 3G,H and EV3E, 500 nM mCherry-p62 S403E, 250 nM NBR1, 250 nM mCherry-TAX1BP1, 250 nM SINTBAD or NAP1, 500 nM GFP-TBK1, 500 nM SINTBAD-GFP were mixed in SEC buffer (25 mM HEPES, pH 7.5, 150 mM NaCl, 1 mM DTT) in a 384-well plate (781892, Greiner Bio-One). In all, 1.5 µM GST-4xUB was added to the wells at the microscope to trigger condensation of the proteins.

For the competition assay with the GABARAP proteins in Fig. 5E, 500 nM mCherry-p62 S403E, 250 nM NBR1, 250 nM GFP-TAX1BP1 and increasing amounts of GABARAP (0 nM–500 nM–1 µM–5 µM) were mixed in SEC buffer (25 mM HEPES, pH 7.5, 150 mM NaCl, 1 mM DTT) in a 384-well plate. 1.5 µM GST-4xUB was added at the microscope to trigger condensation of the proteins.

For the condensation assay in Fig. 7J, 500 nM mCherry-p62 S403E, 250 nM NBR1, 250 nM GFP-TAX1BP1 and either 0 µM or 5 µM GABARAP were mixed in SEC buffer (25 mM HEPES, pH 7.5, 150 mM NaCl, 1 mM DTT) in a 384-well plate. 1.5 µM GST-4xUB or GST-8xUB were added to the wells at the microscope to trigger condensation of the proteins.

Images depicted in Figs. 1E, 2G, 2H, 5E, and 7B show indicated timepoints after subtracting the background using rolling ball background reduction (rolling ball radius = 5 pixels). For quantifications, the same macro was used as previously described (Turco et al, 2021; Zaffagnini et al, 2018). In short, rolling ball background subtraction of ImageJ was performed, followed by thresholding and puncta counting across all channels and timepoints using the Analyze puncta function.

## Peptide array

The peptide array was synthesized by the Peptide Synthesis facility of the IMP (https://cores.imp.ac.at/peptide-synthesis/). The entire NBR1 protein sequence was spotted as 12 aa/spot in a 2 aa window. The membrane was first soaked in methanol and then washed 3× with TBS. The membrane was incubated with blocking buffer (3% BSA dissolved in 1× TBS) overnight at 4 °C. The next day the membrane was washed 1× with TBST (TBS + 0.1% Tween-20) and incubated with our protein (GFP-TAX1BP1) at a concentration of 50 µg/ml in 15 ml of SEC Buffer (25 mM HEPES, pH 7.5, 150 mM NaCl, 1 mM DTT) for 2 h at RT. After the incubation, the membrane was washed 3x with TBST.

For detection, a western blot using a PVDF membrane (88520, Thermo) was performed. After the transfer by wet blotting, the membranes were treated as described above.

## Statistical analysis

For the statistical analysis, Prism GraphPad was used. For all microscopy-based interaction assays a one-way ANOVA was performed. For immunofluorescence images Student's $t$ test (unpaired) was performed. For live-cell imaging data, two-way ANOVA or Student's $t$ test was performed. For western blot quantifications, one-way ANOVAs or Student's $t$ tests (unpaired) were performed. For condensation assays, either one-way ANOVA or Student's $t$ test (unpaired) was performed depending on the number of conditions. For comparing the number of foci between TAX1BP1 WT and ΔZnF we performed a one-sample $t$ and Wilcoxon test. $P$ values were corrected for multiple testing using the suggested method. All values comparing data points in the figures refer to $P$ values. Error bars are reported as mean ± standard deviation.

## Data availability

All reagents, constructs, and cell lines will be available upon request. Source data is provided with this paper.

The source data of this paper are collected in the following database record: biostudies:S-SCDT-10_1038-S44318-024-00280-5.

## Peer review information

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

## Acknowledgements

The authors thank all members of the Martens lab for fruitful discussions and Susanna Tulli and Elias Adriaenssens for comments on the manuscript. The authors also thank Elias Adriaenssens for providing SINTBAD protein, Michael Lazarou for providing the PentaKO HeLa cells, Jean-Marc Furlano for help with the NBR1–TAX1BP1 interaction assays, Lea Radzuweit for help with the purification of mCherry-p62 S403E and the Clausen lab (IMP, Vienna) for help with the peptide array. The authors further thank the Max Perutz Labs BioOptics facility and the Mass Spectrometry facility for technical support. Proteomics analyses were performed using the VBCF instrument pool. The authors also thank the VBCF Protech facility for the HEK cell expressions. This research was funded in whole or in part by the Austrian Science Fund (FWF) (DOI:10.55776/P30401-B21, DOI:10.55776/PAT7165623, DOI:10.55776/W1261 and DOI: 10.55776/F79) and the European Union's Framework Programme for Research and Innovation Horizon 2020 (2014-2020) under the Marie Curie Skłodowska Grant Agreement No. 847548. For open access purposes, the author has applied a CC BY public copyright license to any author accepted manuscript version arising from this submission.

## Author contributions

**Bernd Bauer**: Conceptualization; Data curation; Formal analysis; Supervision; Investigation; Methodology; Writing—original draft; Writing—review and editing. **Jonas Idinger**: Formal analysis; Investigation. **Martina Schuschnig**: Resources; Investigation. **Luca Ferrari**: Conceptualization; Supervision. **Sascha Martens**: Conceptualization; Resources; Supervision; Funding acquisition; Writing—original draft; Project administration; Writing—review and editing.

Source data underlying figure panels in this paper may have individual authorship assigned. Where available, figure panel/source data authorship is listed in the following database record: biostudies:S-SCDT-10_1038-S44318-024-00280-5.

## Disclosure and competing interests statement

Sascha Martens is a member of the scientific advisory board of Casma Therapeutics. The remaining authors declare no competing interests.

# Expanded View Figures

**Figure EV1.  TAX1BP1 is not a constitutive component of p62 condensates in cells.**

(**A**) Western blot for the validation of the CRISPR knock-in cell line (PAR = parental cell line). (**B**) Quantification of (**A**). The increase between untreated and Bafilomycin treatment was plotted and compared to the parental cell line (PAR). (**C**) Representative immunofluorescence images of HAP1 WT + /- VPS34 IN1, ATG14 KO, ATG7 KO and FIP200 KO cells stained for p62 and TAX1BP1 (scale bar = 20 μm). Line scan of fluorescence intensities of p62 and TAX1BP1 to the right side of the corresponding images. The data in (**B**) shows the mean +/− s.d. from three independent experiments. Two-way ANOVA with Sidak's multiple comparison test was performed in (**B**). Source data are available online for this figure.

▶

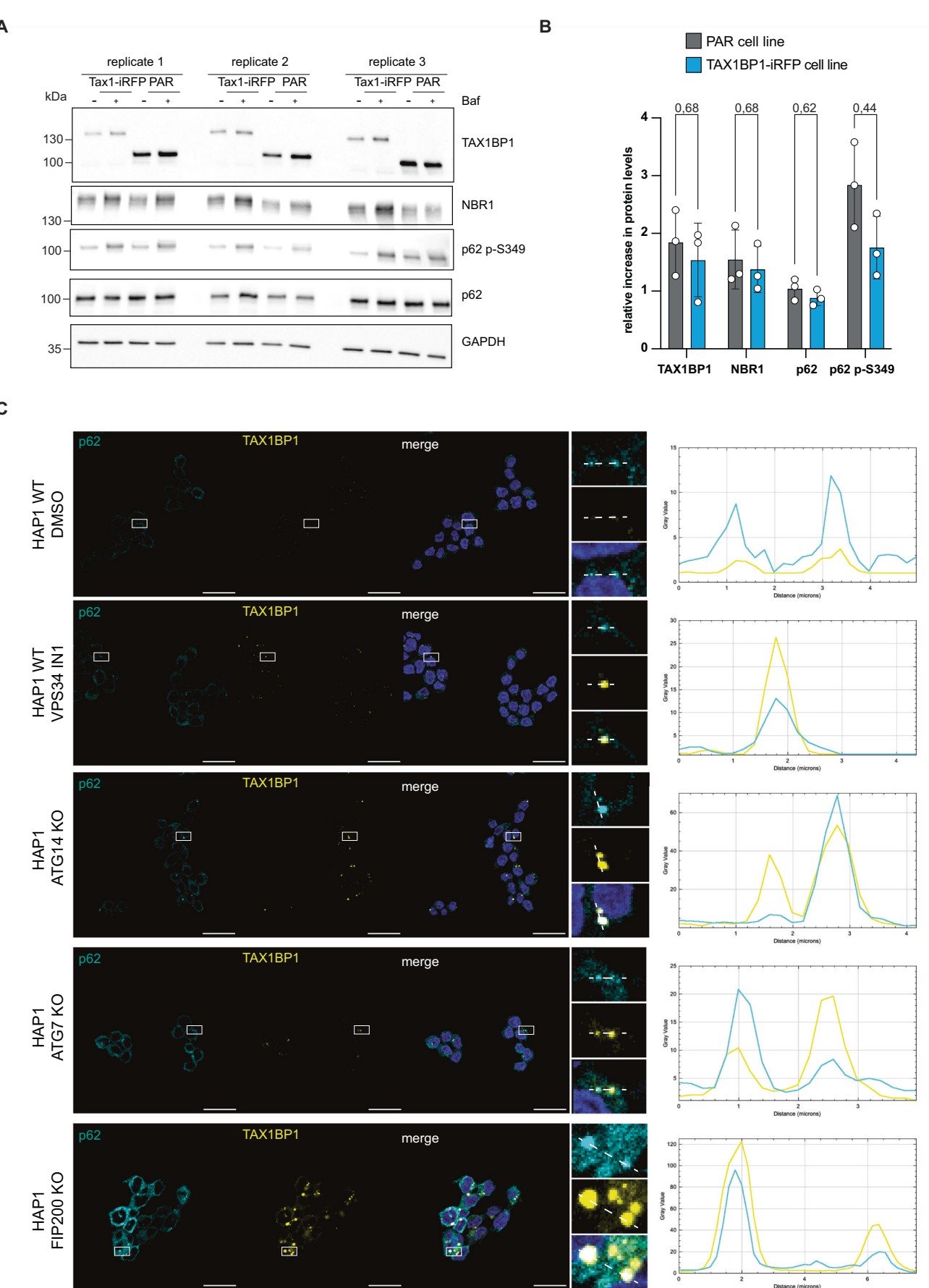

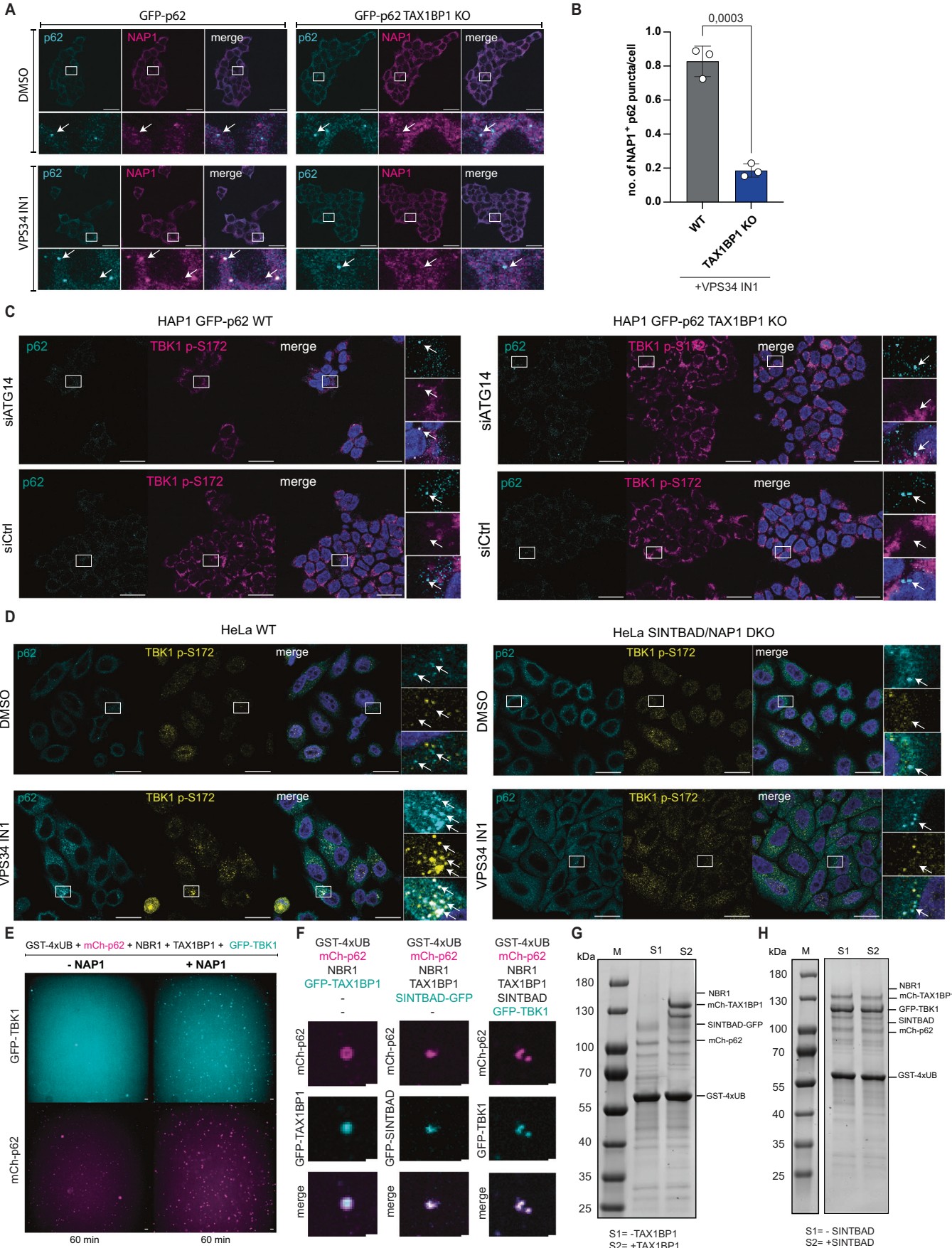

**Figure EV2. TAX1BP1 recruits TBK to p62 condensates via the adapter protein SINTBAD/NAP1.**

(A) Representative immunofluorescence images of cells stained for p62 and NAP1 untreated and upon autophagy inhibition (scale bar = 20 μm). NAP1 colocalization with p62 puncta was abrogated in TAX1BP1 KO cell line (white arrows). (B) Quantification of the number of p62 puncta positive for NAP1 upon autophagy inhibition per cell and focal plane in (A). (C) Representative immunofluorescence images of GFP-p62 WT and TAX1BP1 KO cells stained for TBK1 p-S172 after siRNA-mediated knock-down of ATG14 (scale bar = 20 μm). The TBK1 p-S172 colocalization with p62 puncta was abrogated in TAX1BP1 KO cell line (white arrows). (D) Representative immunofluorescence images of HeLa WT and HeLa DKO (NAP1 & SINTBAD KO) cells stained for p62 and TBK1 p-S172 upon VPS34 IN1 treatment (scale bar = 20 μm). The TBK1 p-S172 colocalization with p62 puncta was abrogated in SINTBAD/NAP1 DKO cell line (white arrows). (E) Representative images of the condensation assay showing NAP1-dependent recruitment of GFP-TBK1 to p62 condensates in vitro (scale bar = 10 μm). (F) Representative images of p62 condensates at a higher magnification (scale bar = 1 μm). (G) SDS Page gel as a loading control for the condensation assay in Fig. 3G. (H) SDS Page gel as a loading control for the condensation assay in 3H. Data in (B) shows the mean +/− s.d. from three independent experiments. Unpaired *t* test was performed in (B). Source data are available online for this figure.

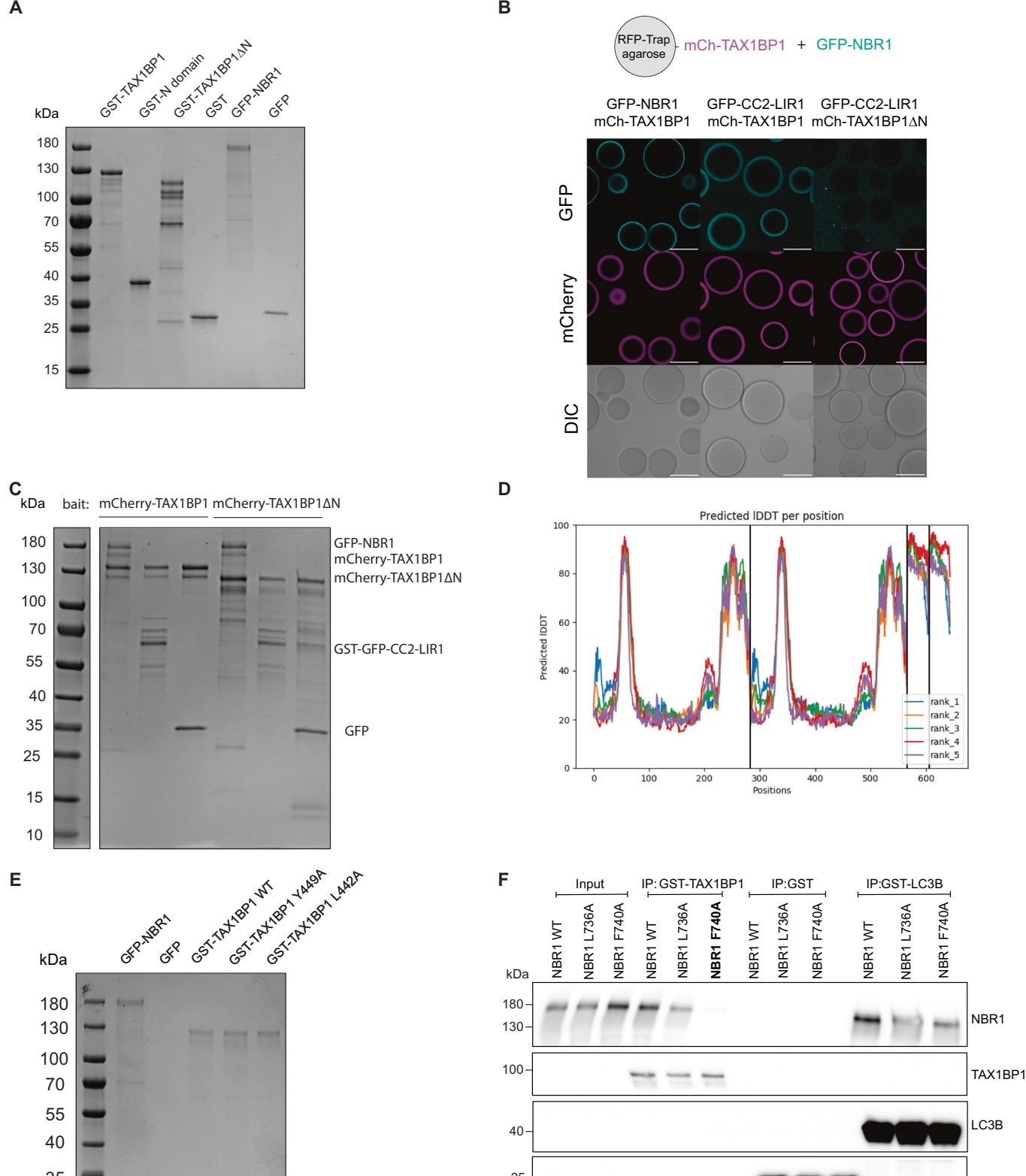

**Figure EV3.  Mapping the interaction between TAX1BP1 and NBR1.**

(**A**) SDS Page gel as loading control for microscopy-based interaction assay in 4 A. (**B**) Representative images for microscopy-based interaction assay with GFP tagged NBR1 fragment (CC2-LIR2) and mCherry-TAX1BP1 FL or ΔN on RFP-trap beads (scale bar = 80 μm). (**C**) SDS Page gel as a loading control for the interaction assay in (**B**). (**D**) pLDDT plot for the AlphaFold 2 prediction in Fig. 4E. (**E**) SDS Page gel as a loading control for the interaction assay in Fig. 4F. (**F**) Western blot of pulldown with cells expressing NBR1 point mutants including GST-LC3B as bait to determine the effect of point mutant on GABARAP and LC3B binding. Source data are available online for this figure.

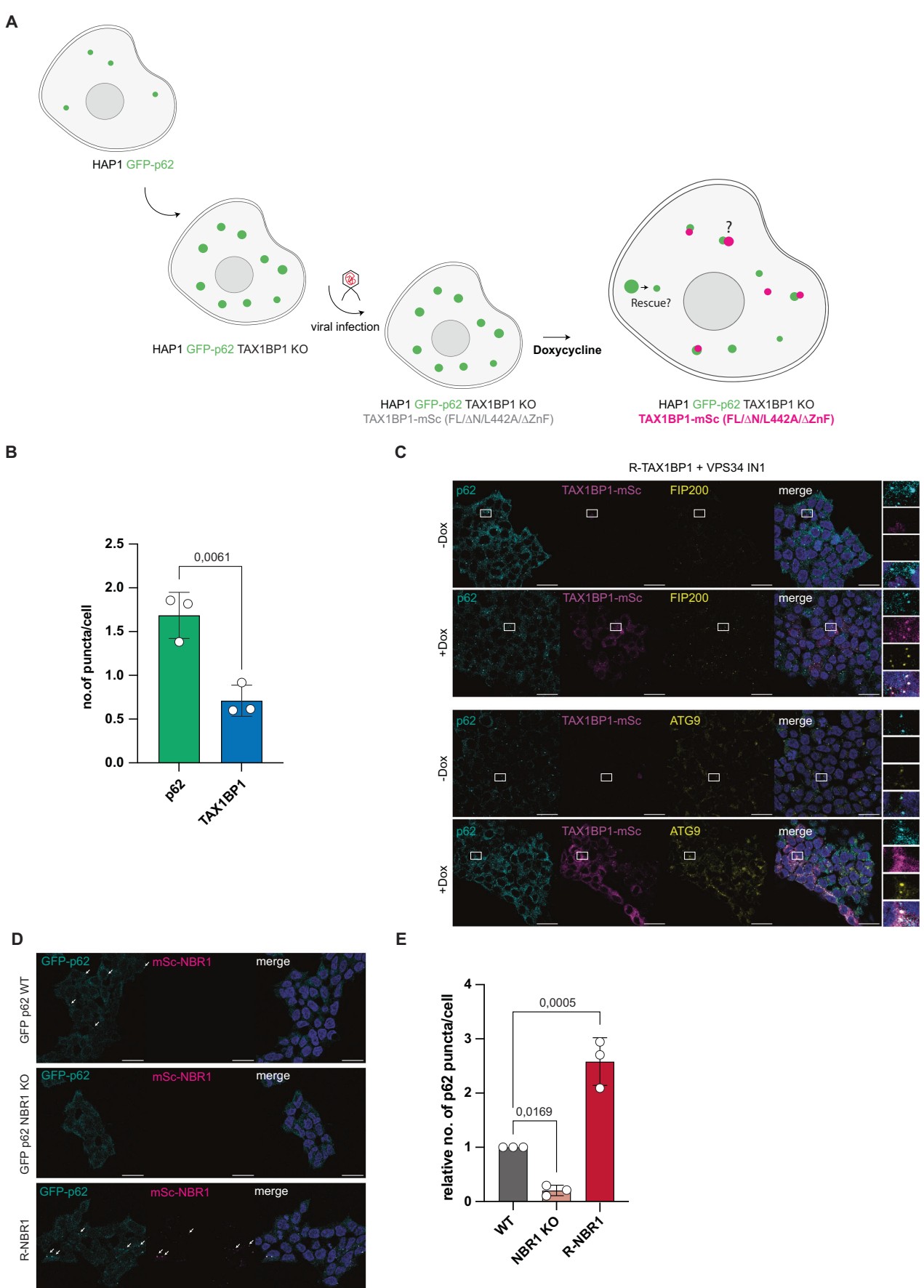

◄  **Figure EV4.   The TAX1BP1–NBR1 interaction is required for the autophagic flux of p62 condensates.**

(A) Schematic overview for the experimental set-up of the experiment shown in Fig. 6. (B) Quantification of the number of p62 and TAX1BP1 foci in R-TAX1BP1 cells upon VPS34 IN1 treatment per cell and focal plane. (C) Representative immunofluorescence images of R-TAX1BP1 cell line treated with VPS34 IN1 and $+/-$ doxycycline to induce re-expression of TAX1BP1-mSc and stained for FIP200 and ATG9 (scale bar $=$ 20 μm). (D) Representative immunofluorescence images of GFP-p62 WT, NBR1 KO and NBR1 KO cells re-expressing mSc-NBR1 (scale bar $=$ 20 μm). p62 & mSc-NBR1 puncta are highlighted (white arrows). (E) Quantification of number of p62 puncta per cell and focal plane in (D). The number of p62 puncta was plotted as a relative to GFP-p62 WT. Data in (B, E) shows the mean $+/-$ s.d. from three independent experiments. Unpaired *t* test was performed in (B). One-way ANOVA with Dunnett's multiple comparison test was performed in (D). Source data are available online for this figure.

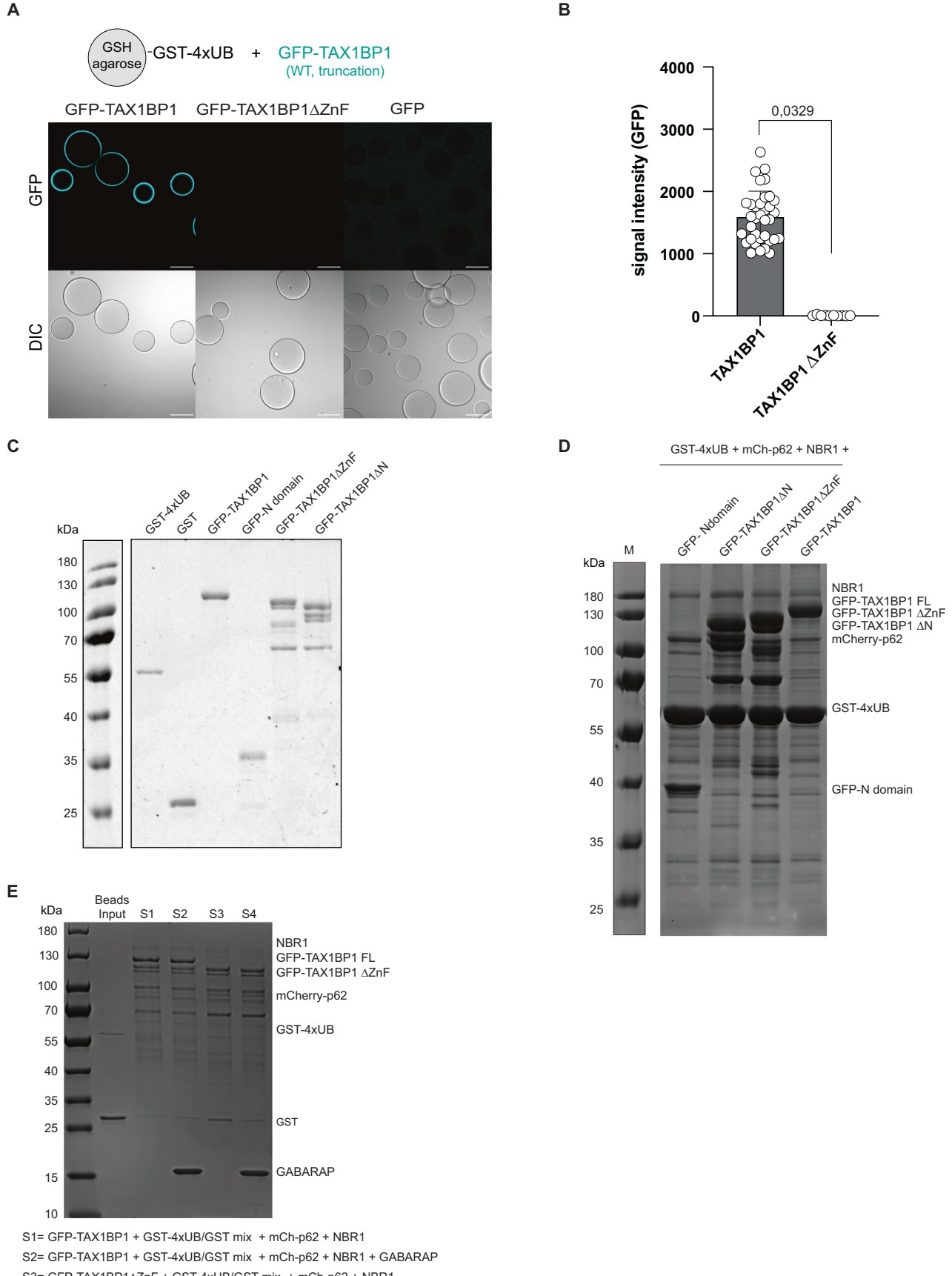

S1= GFP-TAX1BP1 + GST-4xUB/GST mix + mCh-p62 + NBR1
S2= GFP-TAX1BP1 + GST-4xUB/GST mix + mCh-p62 + NBR1 + GABARAP
S3= GFP-TAX1BP1ΔZnF + GST-4xUB/GST mix + mCh-p62 + NBR1
S4= GFP-TAX1BP1ΔZnF + GST-4xUB/GST mix + mCh-p62 + NBR1 + GABARAP

◀  **Figure EV5.   Ubiquitin interaction stabilizes the recruitment of TAX1BP1.**

(**A**) Representative images of a microscopy-based protein–protein interaction assay using the indicated proteins (scale bar = 80 μm). (**B**) Quantification of signal intensity of GFP-TAX1BP1 and GFP-TAX1BP1ΔZnF on GST-4xUB coated beads in (**A**). (**C**) SDS Page gel as a loading control for the microscopy-based interaction assay shown in (**A**). (**D**) SDS Page gel as a loading control for the condensation assay shown in Fig. 7B. (**E**) SDS Page gel as a loading control for the microscopy-based interaction assay in Fig. 7I. The data in (**B**) shows the mean +/- s.d. from three independent experiments. Each data point in (**B**) represents the mean signal intensity for an individual bead. Background-corrected values were used for statistical analysis and negative values were excluded. Nested one-way ANOVA with Dunnett's multiple comparison test was performed in (**B**). Source data are available online for this figure.

