## [Peer Review File · The EMBO Journal]

Recruitment of autophagy initiator TAX1BP1 advances aggrephagy from cargo collection to sequestration

Bernd Bauer, Jonas Idinger, Martina Schuschnig, Luca Ferrari, and Sascha Martens

Corresponding author(s): Sascha Martens (sascha.martens@univie.ac.at) , Bernd Bauer (bernd.bauer@univie.ac.at)

Review Timeline:

Submission Date:	17th May 24
Editorial Decision:	20th Jun 24
Revision Received:	5th Sep 24
Editorial Decision:	27th Sep 24
Revision Received:	1st Oct 24
Accepted:	8th Oct 24

Editor: William Teale

Transaction Report:

Dear Sascha,

Thank you again for the submission of your manuscript entitled "Mechanism of TAX1BP1 recruitment in aggrephagy to switch from cargo collection to sequestration" . We have now received the reports from the referees, which I copy below.

As you can see from their comments, they found your study timely and considered the experiments well designed and carefully conducted. Recommendations have been made for how the study could be further improved which I would like you to consider carefully. Some of the points will certainly require your attention before your manuscript can be published in The EMBO Journal.

Based on the overall interest expressed in the reports, I would like to invite you to address the comments of all referees in a revised version of the manuscript. I should add that it is The EMBO Journal policy to allow only a single major round of revision and that it is therefore important to resolve the main concerns at this stage. I believe the points that the referees make are reasonable and addressable, but please contact me if you have any questions, need further input on the referee comments or if you anticipate any problems. Once you have had a chance to digest the reports, I suggest we meet by Zoom to discuss the revisions you propose. Please, follow the instructions below when preparing your manuscript for resubmission.

I would also like to point out that as a matter of policy, competing manuscripts published during this period will not be taken into consideration in our assessment of the novelty presented by your study ("scooping" protection). We have extended this 'scooping protection policy' beyond the usual 3 month revision timeline to cover the period required for a full revision to address the essential experimental issues. Please contact me if you see a paper with related content published elsewhere to discuss the appropriate course of action.

Again, please contact me at any time during revision if you need any help or have further questions.

Thank you very much again for the opportunity to consider your work for publication. I look forward to your revision.

Best regards,

William

William Teale, Ph.D.
Editor
The EMBO Journal

When submitting your revised manuscript, please carefully review the instructions below and include the following items:

- 1) a .docx formatted version of the manuscript text (including legends for main figures, EV figures and tables). Please make sure that the changes are highlighted to be clearly visible.
- 2) individual production quality figure files as .eps, .tif, .jpg (one file per figure).
- 3) a .docx formatted letter INCLUDING the reviewers' reports and your detailed point-by-point response to their comments. As part of the EMBO Press transparent editorial process, the point-by-point response is part of the Review Process File (RPF), which will be published alongside your paper.
- 4) a complete author checklist, which you can download from our author guidelines ([https://wol-prod-cdn.literatumonline.com/pb-assets/embo-site/Author Checklist%20-%20EMBO%20J-1561436015657.xlsx](https://wol-prod-cdn.literatumonline.com/pb-assets/embo-site/Author%20Checklist%20-%20EMBO%20J-1561436015657.xlsx)). Please insert information in the checklist that is also reflected in the manuscript. The completed author checklist will also be part of the RPF.
- 5) Please note that all corresponding authors are required to supply an ORCID ID for their name upon submission of a revised manuscript.
- 6) We require a 'Data Availability' section after the Materials and Methods. Before submitting your revision, primary datasets produced in this study need to be deposited in an appropriate public database, and the accession numbers and database listed under 'Data Availability'. Please remember to provide a reviewer password if the datasets are not yet public (see <https://www.embopress.org/page/journal/14602075/authorguide#datadeposition>). If no data deposition in external databases is

needed for this paper, please then state in this section: This study includes no data deposited in external repositories. Note that the Data Availability Section is restricted to new primary data that are part of this study.

Note - All links should resolve to a page where the data can be accessed.

8) For data quantification: please specify the name of the statistical test used to generate error bars and P values, the number (n) of independent experiments (specify technical or biological replicates) underlying each data point and the test used to calculate p-values in each figure legend. The figure legends should contain a basic description of n, P and the test applied. Graphs must include a description of the bars and the error bars (s.d., s.e.m.).

9) We would also encourage you to include the source data for figure panels that show essential data. Numerical data can be provided as individual .xls or .csv files (including a tab describing the data). For 'blots' or microscopy, uncropped images should be submitted (using a zip archive or a single pdf per main figure if multiple images need to be supplied for one panel). Additional information on source data and instruction on how to label the files are available at .

10) We replaced Supplementary Information with Expanded View (EV) Figures and Tables that are collapsible/expandable online (see examples in <https://www.embopress.org/doi/10.15252/embj.201695874>). A maximum of 5 EV Figures can be typeset. EV Figures should be cited as 'Figure EV1, Figure EV2" etc. in the text and their respective legends should be included in the main text after the legends of regular figures.

12) Our journal encourages inclusion of *data citations in the reference list* to directly cite datasets that were re-used and obtained from public databases. Data citations in the article text are distinct from normal bibliographical citations and should directly link to the database records from which the data can be accessed. In the main text, data citations are formatted as follows: "Data ref: Smith et al, 2001" or "Data ref: NCBI Sequence Read Archive PRJNA342805, 2017". In the Reference list, data citations must be labeled with "[DATASET]". A data reference must provide the database name, accession number/identifiers and a resolvable link to the landing page from which the data can be accessed at the end of the reference. Further instructions are available at .

We realize that it is difficult to revise to a specific deadline. In the interest of protecting the conceptual advance provided by the work, we recommend a revision within 3 months (18th Sep 2024). Please discuss the revision progress ahead of this time with the editor if you require more time to complete the revisions. Use the link below to submit your revision:

Referee #1:

This manuscript shows the molecular mechanism of TAX1BP1 recruitment to p62 condensates during autophagosome biogenesis. The authors reveal that TAX1BP1 is recruited to the condensates through the interaction with NBR1 and ubiquitin while TAX1BP1 is not constitutive component of the condensates. The author also reveals that TAX1BP1 recruits TBK1 via SINTBAD. From these results, the authors demonstrate a novel role for TAX1BP1 in TBK1 recruitment via the NBR1-TAX1BP1-NAP1/SINTBAD axis. Furthermore, the model is proposed that TAX1BP1 is recruited to p62 condensates in dependent on ubiquitin load, based on the finding that GABARAP competes with TAX1BP1 for NBR1 binding. The experiments are well-designed and the results are largely convincing. Their findings advance our understanding of the molecular consequences of autophagosome biogenesis in aggrephagy. However, there are several key points that require further clarification.

Major comments

1. In this manuscript, the authors propose that TBK1 recruitment to p62 condensates requires TBK1 adaptor protein, SINTBAD and NAP1. However, the relative roles of these adaptor proteins remain unclear.
 - Does TAX1BP1 knockout affect the colocalization of SINTBAD/NAP1 with p62 in cells?
 - In Fig. 3F-K, the authors show that TBK1 is recruited to p62 condensates through TAX1BP1 and SINTBAD. Is NAP1 also essential for TBK1 recruitment in vitro assay?
 - The argument that SINTBAD/NAP1 is essential for the TBK1 recruitment to p62 condensates is not demonstrated from in vivo experiment using cells. To support the model, functional assay using cell should be included.
2. The manuscript suggests that GABARAP competes with TAX1BP1 for NBR1 binding. In vivo confirmation is necessary.
 - Are TAX1BP1 signals are rarely detected on GABARAP/LC3-positive NBR1 puncta in cells?
 - Does GABARAP/LC3 binding with p62 condensates decrease upon TAX1BP1 inclusion in the in vitro assay?
3. Does NBR1 knockout affect the recruitment of TAX1BP1 to p62 condensates in cells?
It is unclear whether reintroduction of NBR1 F740A mutant in NBR1 knockout cells fails to rescue the recruitment of TAX1BP1 to p62 condensates.
4. Regarding NAP1 phosphorylation, the lower level observed in TAX1BP1-mSc expressing TAX1BP1 knockout cells suggests a partial functional defect in TAX1BP1-mSc.

Minor comments

1. Fig7. C: Please include data for p62 pS403 signals as shown in Fig. 3B and Fig. 6E.

Referee #2:

The work entitled "Mechanism of Taxbp1 Recruitment in Aggrephagy to Switch from Cargo Collection to Sequestration" by Sascha Martens's group adds additional molecular details to the aggrephagy mechanism, a process whereby ubiquitinated protein condensates are degraded via autophagy. The authors demonstrate that the recruitment of TAXBP1 to P62 ubiquitinated cargo occurs dynamically and represents the initiation of autophagosome biogenesis and recruitment of the selective autophagy kinase TBK1 via the presence of the SINTBAD adapter. They characterize the minimal region of the TAXBP1-NBR1 binding site and demonstrate that TAXBP1 competes with GABARAP for NBR1 binding. The N domain of TAXBP1 that binds to the LIR domain of NBR1 is required for the binding of TAXBP1 to ubiquitin condensates decorated with P62. The ubiquitin-binding domain of TAXBP1, on the other hand, seems important to maintain TAXBP1 association with ubiquitin condensates in the presence of GABARAP.

The work is clearly written, and the data are of high quality. In general, the conclusions are supported by the data. However, I have a few questions that the authors should try to address (if feasible).

It is unclear to me what determines the arrival of TAXBP1 to P62-condensate. Is this dependent on NBR1 levels, or is there a ubiquitin threshold? Alternatively, does the binding of P62 to GABARAP/LC3 prevent the arrival of TAXBP1?

I suggest the authors perform additional experiments in cellular models to support some of the most intriguing results obtained using recombinant proteins in vitro. For example, it would be interesting to study the competition between GABARAP and TAXBP1 binding to NBR1 in cells that overexpress either one of the two proteins. How will aggrephagy flux be affected?

The hypothesis that a ubiquitin threshold can regulate TAXBP1 recruitment to P62 is interesting, but it is not yet supported by the presented data. Can the authors investigate this aspect further, for example by modulating the levels of ubiquitinated proteins?

Does the TAXBP1 N-region bind other autophagy receptors via the LIR domain? Could this be a generalized mechanism to bind other autophagy receptors, or is it specific to NBR1?

Referee #3:

Autophagic clearance of protein aggregates is a pivotal topic in the field of autophagy. Various pathways for aggregate clearance have been documented, contingent on the properties of the aggregates. Among these pathways, the ubiquitin-dependent clearance of P62 condensates is extensively studied. However, the intricate interactions among multiple ubiquitin-binding receptors in this process remain unresolved. In the present study, the authors elucidate a critical interaction between NBR1 and TAX1BP1, essential for the autophagic clearance of P62. The study is conducted with the same level of sophistication as the senior author's previous research. Below are some comments for further refinement of the study:

1. ****Introduction****: It would be beneficial to briefly outline the multiple pathways of aggrephagy, including the clearance of liquid P62 condensates, gel aggregate clearance in *C. elegans*, cytoplasm-to-vacuole targeting (Cvt), and solid aggregate clearance.
2. ****Figure 1A****: The authors demonstrated enhanced colocalization between TAX1BP1 and P62 condensates with PI3K inhibitors. It would be advantageous to confirm this finding using Atg5KO and Atg14KO models or RNAi to ascertain the dependence on autophagic factors. Similar validations are also required for Figures 2E, 3D, 6A, and 7C.
3. ****Figure 2C and D****: There is a discrepancy between the quantification and the Western blot (WB) data. Specifically, KO2 exhibited more NBR1 in the WB but less in the quantification.
4. ****Figure 2A and E****: The effect of TAX1BP1 deficiency on P62 autophagy should be validated by reexpression of TAX1BP1.
5. ****Figure 3F-H****: It would be helpful to magnify the phase-separated droplets and determine the localization of TAX1BP1, SINTBAD, and TBK1 within the droplets. Are these proteins uniformly distributed or localized on the surface?
6. ****Figure 6D****: To substantiate the effect of TAX1BP1 binding on P62 clearance, autophagic flux assays are needed to confirm the turnover of P62. Additionally, this should be validated by depleting ATG5 and ATG14 alongside PI3K inhibitors.
7. ****Figure 5****: The authors need to clarify the functional role underlying the competition between GABARAP and TAX1BP1 for

NBR1 binding.

8. **Figure S4F**: The gel should be replaced with a high-quality blot to ensure clarity and accuracy.

Referee #1:

This manuscript shows the molecular mechanism of TAX1BP1 recruitment to p62 condensation during autophagosome biogenesis. The authors reveal that TAX1BP1 is recruited to the condensates through the interaction with NBR1 and ubiquitin while TAX1BP1 is not constitutive component of the condensates. The author also reveals that TAX1BP1 recruits TBK1 via SINTBAD. From these results, the authors demonstrate a novel role for TAX1BP1 in TBK1 recruitment via the NBR1-TAX1BP1-NAP1/SINTBAD axis. Furthermore, the model is proposed that TAX1BP1 is recruited to p62 condensates in dependent on ubiquitin load, based on the finding that GABARAP competes with TAX1BP1 for NBR1 binding.

The experiments are well-designed and the results are largely convincing. Their findings advance our understanding of the molecular consequences of autophagosome biogenesis in aggrephagy. However, there are several key points that require further clarification.

We are grateful to the reviewer's positive assessment of our work and the constructive comments.

Major comments

1. In this manuscript, the authors propose that TBK1 recruitment to p62 condensates requires TBK1 adaptor protein, SINTBAD and NAP1. However, the relative roles of these adaptor proteins remain unclear.

- Does TAX1BP1 knockout affect the colocalization of SINTBAD/NAP1 with p62 in cells?

We apologize for not being sufficiently clear in the original draft of the manuscript. We performed immunofluorescence (IF) for NAP1 in WT and TAX1BP1 KO cells expressing GFP-p62. We found a significant reduction in the colocalization of NAP1 with GFP-p62 in the TAX1BP1 KO cells in comparison to the WT cells (see Fig. EV3A, B).

- In Fig. 3F-K, the authors show that TBK1 is recruited to p62 condensates through TAX1BP1 and SINTBAD. Is NAP1 also essential for TBK1 recruitment in vitro assay?

We thank the reviewer for raising this point. In response, we performed the condensation assay also with NAP1 and found a similar dependency for TBK1 recruitment as with SINTBAD (Fig. EV3E).

- The argument that SINTBAD/NAP1 is essential for the TBK1 recruitment to p62 condensates is not demonstrated from in vivo experiment using cells. To support the model, functional assay using cell should be included.

For the revised manuscript, we performed IF for p62 and TBK1 in HeLa WT cells and HeLa DKO cells (lacking NAP1 & SINTBAD). There was a clear loss of colocalization in the DKO cell line in comparison to the WT cell line (Fig. EV3D).

2. The manuscript suggests that GABARAP competes with TAX1BP1 for NBR1 binding. In vivo confirmation is necessary

- Are TAX1BP1 signals are rarely detected on GABARAP/LC3-positive NBR1 puncta in cells?

This is indeed a very good point, and we carefully considered how best to perform an experiment testing this hypothesis. TAX1BP1 also binds to GABARAP/LC3 proteins and p62 as a major component of the condensates also binds these proteins. In addition, some of these condensates may be associated with phagophores or sequestered in autophagosomes, which are also positive for GABARAP/LC3. It is therefore not straightforward to correlate the NBR1 – TAX1BP1 interaction with the intensity of

GABARAP/LC3. For this reason, we conducted a slightly different experiment. We generated a NBR1 KO cell line and stably re-expressed NBR1 WT and NBR1 mutLIR1 (deficient in GABARAP/LC3 binding). We detected a significant increase in the levels of p62 phosphorylated at S403 in the NBR1 mutLIR1 expressing cells (Fig. RL1A, B). Since we have shown in the manuscript that TAX1BP1 is the main recruiter of TBK1, which phosphorylates p62 at S403, this suggests that TAX1BP1 is more robustly recruited in these cells. In addition, TAX1BP1 levels are a little bit lower in the NBR1 mutLIR1 and are stabilized to a higher degree upon block of autophagy by VPS34 IN1 treatment (Fig. RL1A, B). This would speak for increased p62 activation and autophagic flux in the NBR1 mutLIR1 compared to the NBR1 WT expressing cells. We also performed IF using the same cells to compare the number of p62 puncta between WT and mutLIR1. In this experiment, there was no large difference in the total number of p62 puncta neither in untreated cells nor in cells treated with VPS34 IN1 (not shown), but p62 tended to accumulate more in the NBR1 mutLIR1 cells upon VPS34 IN1 treatment (Fig. RL1C).

Our data indicate that there is a tendency for faster flux of p62 condensates in the NBR1 mutLIR1 expressing cells, consistent with the hypothesis that TAX1BP1 is more robustly recruited. This is noteworthy because canonically it would have been assumed that reducing the ability of NBR1 to bind GABARAP/LC3 decreases autophagic flux. However, since the effects are small, the exact expression levels of NBR1 difficult to control and the results somewhat variable, we are hesitant to add these data to the manuscript. Nonetheless, we fully agree that this should be followed up in a more controllable system.

We also addressed the role of the competition between GABARAPs and TAX1BP1 in a more complex in vitro setting. To this end, we performed condensation assays with longer ubiquitin chains to mimic a cargo with a higher ubiquitin load and indeed we found that TAX1BP1 was recruited to p62 condensates even in the presence of GABARAP proteins. (see Fig. 7J, K)

Fig. RL1A) Western blots comparing a NBR1 KO cell line re-expressing NBR1 WT and NBR1 mutLIR1 in the presence and absence of VPS34 IN1 treatment. B) Quantification of phosphorylation levels of p62 and total TAX1BP1 protein levels from three western blots, one of which is shown in A. TAX1BP1 and p62 phosphorylation levels were quantified as a relative to R-NBR1. C) Quantification of IF experiments. The change in the number of p62 condensates upon VPS34 IN1 treatment is compared between NBR1 KO cells re-expressing NBR1 WT or NBR1 mutLIR1. Data in B and C show the mean +/- s.d. from three independent experiments. Two-way ANOVA with Sidak's multiple comparison test was performed in B. Unpaired t-test was performed in C., **P<0.005.

- Does GABARAP/LC3 binding with p62 condensates decrease upon TAX1BP1 inclusion in the in vitro assay?

We thank the reviewer for the excellent suggestion. We tested this in our condensation assay. However, the experiment is difficult to interpret. First, due to the high mobility of the condensates in solution and technical limitations of the available spinning disc system, we are unable to image p62 in addition to GABARAP and TAX1BP1. Thus, in this particular experiment we are blind to the number and size of p62 condensates. In addition, while TAX1BP1 likely competes with GABARAP for NBR1 binding, it also brings along an additional LIR motif. Furthermore, the LIR motif of p62 (the most abundant component in the condensates) can still bind to GABARAP.

In the experiment shown below, it seems as if GABARAP is recruited a bit more efficiently but for the reasons mentioned above, we refrained from including it in the manuscript.

Fig. RL2) Representative image of a condensation assay mixing GST-4xUB, Halo-p62, NBR1, GFP-TAX1BP1 and mCh-GABARAP in a 384 well glass-bottom plate imaged by spinning disc microscopy.

3. Does NBR1 knockout affect the recruitment of TAX1BP1 to p62 condensates in cells? It is unclear whether reintroduction of NBR1 F740A mutant in NBR1 knockout cells fails to rescue the recruitment of TAX1BP1 to p62 condensates.

For the revised manuscript, we first generated a NBR1 KO cell line. This cell line recapitulated previous reports showing that NBR1 promotes the formation of p62 condensates (Fig. EV6D, E).

We then re-expressed the mutant and WT version of NBR1 in this KO cell line and found that, consistent with the results shown in Fig. 4H, the F740A mutation led to a loss of TAX1BP1 recruitment to p62 condensates and to an increased number of p62 condensates in comparison to the WT (see Fig. 6H-J).

4. Regarding NAP1 phosphorylation, the lower level observed in TAX1BP1-mSc expressing TAX1BP1 knockout cells suggests a partial functional defect in TAX1BP1-mSc.

Yes, we also noticed a difference between the rescue cell line and the parental cell. We are confident, however, that the reason is not a functional defect but rather due to a subpopulation of cells that doesn't express TAX1BP1. The cells were not sorted after viral transductions and selected only with G418 which is not the most stringent selection marker. Therefore, we believe that there might be a TAX1BP1⁻ subpopulation that leads to the lower signals. For this reason, we also always compared the mutants to the R-TAX1BP1 WT cell line.

Minor comments

1. Fig7. C: Please include data for p62 pS403 signals as shown in Fig. 3B and Fig. 6E. The quantification of p62 pS403 is now included in Fig. 7H.

Referee #2:

The work entitled "Mechanism of Taxbp1 Recruitment in Aggrephagy to Switch from Cargo Collection to Sequestration" by Sascha Martens's group adds additional molecular details to the aggrephagy mechanism, a process whereby ubiquitinated protein condensates are degraded via autophagy. The authors demonstrate that the recruitment of TAXBP1 to P62 ubiquitinated cargo occurs dynamically and represents the initiation of autophagosome biogenesis and recruitment of the selective autophagy kinase TBK1 via the presence of the SINTBAD adapter. They characterize the minimal region of the TAXBP1-NBR1 binding site and demonstrate that TAXBP1 competes with GABARAP for NBR1 binding. The N domain of TAXBP1 that binds to the LIR domain of NBR1 is required for the binding of TAXBP1 to ubiquitin condensates decorated with P62. The ubiquitin-binding domain of TAXBP1, on the other hand, seems important to maintain TAXBP1 association with ubiquitin condensates in the presence of GABARAP.

The work is clearly written, and the data are of high quality. In general, the conclusions are supported by the data. However, I have a few questions that the authors should try to address (if feasible).

We thank the reviewer for the very kind and positive feedback on our manuscript.

It is unclear to me what determines the arrival of TAXBP1 to P62-condensate. Is this dependent on NBR1 levels, or is there a ubiquitin threshold? Alternatively, does the binding of P62 to GABARAP/LC3 prevent the arrival of TAXBP1?

I suggest the authors perform additional experiments in cellular models to support some of the most intriguing results obtained using recombinant proteins in vitro. For example, it would be interesting to study the competition between GABARAP and TAXBP1 binding to NBR1 in cells that overexpress either one of the two proteins. How will aggrephagy flux be affected?

This is indeed a very good point, and we carefully considered how best to perform an experiment testing this hypothesis. We have overexpressed GABARAP in cells but there was a huge heterogeneity in the expression levels making the experiment impossible to quantify. We expected similar effects for TAX1BP1. For this reason, we conducted a slightly different experiment. We generated a NBR1 KO cell line and stably re-expressed NBR1 WT and NBR1 mutLIR1 (deficient in GABARAP/LC3 binding). We detected a significant increase in the levels of p62 phosphorylated at S403 in the NBR1 mutLIR1 expressing cells (Fig. RL3A, B). Since we have shown in the manuscript that TAX1BP1 is the main recruiter of TBK1, which phosphorylates p62 at S403, this suggests that TAX1BP1 is more robustly recruited in these cells. In addition, TAX1BP1 levels are a little bit lower in the NBR1 mutLIR1 and are stabilized to a higher degree upon block of autophagy by VPS34 IN1 treatment (Fig. RL3A, B). This would speak for increased p62 activation and autophagic flux in the NBR1 mutLIR1 compared to the NBR1 WT expressing cells. We also performed IF using the same cells to compare the number of p62 puncta between WT and mutLIR1. In this experiment, there was no large difference in the total number of p62 puncta neither in untreated cells nor in cells treated with VPS34 IN1 (not shown) but p62 tended to accumulate more in the NBR1 mutLIR1 cells upon VPS34 IN1 treatment (Fig. RL3C). Our data indicate that there is a tendency for faster flux of p62 condensates in the NBR1 mutLIR1 expressing cells, consistent with the hypothesis that TAX1BP1 is more robustly recruited. This is noteworthy because canonically it would have been assumed that reducing the ability of NBR1 to bind GABARAP/LC3 decreases autophagic flux. However, since the effects are small, the exact expression levels of NBR1 difficult to control and the results somewhat variable, we are hesitant to add these data to the manuscript. Nonetheless, we fully agree that this should be followed up in a more controllable system.

Fig. 3A) Western blots comparing a NBR1 KO cell line re-expressing NBR1 WT and NBR1 mutLIR1 in the presence and absence of VPS34 IN1 treatment. B) Quantification of phosphorylation levels of p62 and total TAX1BP1 protein levels from western blots, one of which is shown in A. TAX1BP1 and p62 phosphorylation levels were quantified as a relative to R-NBR1. C) Quantification of IF experiments. The change in the number of p62 condensates upon VPS34 IN1 treatment is compared between NBR1 KO cells re-expressing NBR1 WT or NBR1 mutLIR1. Data in B and C show the mean \pm s.d. from three independent experiments. Two-way ANOVA with Sidak's multiple comparison test was performed in B. Unpaired t-test was performed in C., ** $P < 0.005$.

The hypothesis that a ubiquitin threshold can regulate TAXBP1 recruitment to P62 is interesting, but it is not yet supported by the presented data. Can the authors investigate this aspect further, for example by modulating the levels of ubiquitinated proteins?

We thank the reviewer for this excellent suggestion. Indeed, we performed the condensation assay with longer ubiquitin chains (GST-8xUB instead of GST-4xUB). Consistently, we found that the increase in ubiquitin density on the substrate allowed TAX1BP1 to overcome the competition by GABARAP and the levels of TAX1BP1 recruitment was similar to the level of recruitment to condensates formed with GST-4xUB in the absence of GABARAP (see Fig. 7J, K)

Does the TAXBP1 N-region bind other autophagy receptors via the LIR domain? Could this be a generalized mechanism to bind other autophagy receptors, or is it specific to NBR1?

This is an excellent suggestion; thus, we tested all the soluble cargo receptors we have to our disposal (p62, NBR1, NDP52, OPTN). However, none of them showed specific binding to the N domain, at least under the condition tested and in the absence of any modifications. We show the result of this experiment but have refrained from including it in the manuscript. However, we think this point should be addressed in the future.

Fig. RL4) Representative images of a microscopy-based protein interaction assay. GST-N domain was bound to GSH beads and incubated with four different soluble cargo receptors as preys. The beads were imaged using a LSM700 confocal microscope (scale bar = 80 μ m)

Referee #3:

Autophagic clearance of protein aggregates is a pivotal topic in the field of autophagy. Various pathways for aggregate clearance have been documented, contingent on the properties of the aggregates. Among these pathways, the ubiquitin-dependent clearance of P62 condensates is extensively studied. However, the intricate interactions among multiple ubiquitin-binding receptors in this process remain unresolved. In the present study, the authors elucidate a critical interaction between NBR1 and TAX1BP1, essential for the autophagic clearance of P62. The study is conducted with the same level of sophistication as the senior author's previous research. Below are some comments for further refinement of the study:

We are very grateful for the kind and positive feedback on our manuscript.

1. **Introduction**: It would be beneficial to briefly outline the multiple pathways of aggregate clearance, including the clearance of liquid P62 condensates, gel aggregate clearance in *C. elegans*, cytoplasm-to-vacuole targeting (Cvt), and solid aggregate clearance.

We agree and have added a paragraph to the introduction (second paragraph from top).

2. **Figure 1A**: The authors demonstrated enhanced colocalization between TAX1BP1 and P62 condensates with PI3K inhibitors. It would be advantageous to confirm this finding using Atg5KO and Atg14KO models or RNAi to ascertain the dependence on autophagic factors. Similar validations are also required for Figures 2E, 3D, 6A, and 7C.

We thank the reviewer for raising this excellent point. We performed IF for p62 and TAX1BP1 in WT +/- VPS34 IN1, ATG14 KO, FIP200 KO and ATG7 KO cells. Consistent with the VPS34 IN1 treatment, TAX1BP1 clearly accumulated at p62 condensates whenever the autophagy machinery was blocked or inhibited (see Fig. EV1C).

We also performed a knockdown of ATG14 and found that ATG9 accumulates at p62 condensates in WT but not in the TAX1BP1 KO cell line, consistent with the hypothesis that TAX1BP1 is the main recruiter of the autophagy machinery (see Fig. EV2B).

3. **Figure 2C and D**: There is a discrepancy between the quantification and the Western blot (WB) data. Specifically, KO2 exhibited more NBR1 in the WB but less in the quantification.

We thank the reviewer for spotting the discrepancy. We repeated and exchanged the blot. To avoid any confusion, we only show the untreated samples – as these were also used for the quantification (see Fig 2C, D).

4. **Figure 2A and E**: The effect of TAX1BP1 deficiency on P62 autophagy should be validated by re-expression of TAX1BP1.

We apologize if it was unclear. This experiment is shown later in the manuscript when we analyzed the number of p62 condensates for the TAX1BP1 mutants. We found that while the cell line re-expressing the WT TAX1BP1 had on average 0.5 puncta per cell in a single focal plane – the mutants had consistently with the KO twice the amount (see Fig. 6C).

We also performed IF with the TAX1BP1 WT rescue cell line and found that both ATG9 and FIP200 were robustly recruited to p62 condensates in +DOX treated cells vs -DOX treated cells upon autophagy inhibition (see Fig. S6C).

We performed similar experiments with a NBR1 KO cell line, rescued with WT and the NBR1 F740A point mutant, which is deficient in TAX1BP1 binding. Consistently, we found that the point mutant increased the number of p62 condensates irrespective of VPS34 IN1 treatment, in contrast to the WT (see Fig. 6H-J and Fig. EV6D, E).

5. **Figure 3F-H**: It would be helpful to magnify the phase-separated droplets and determine the localization of TAX1BP1, SINTBAD, and TBK1 within the droplets. Are these proteins uniformly distributed or localized on the surface?

We thank the reviewer for this interesting point. We imaged the condensate at a LSM700 confocal microscope using a higher magnification than for the standard analysis. While TAX1BP1 occasionally appeared to be more central within the condensates when compared to p62, no specific distribution of SINTBAD and TBK1 could be observed under the conditions tested (Fig. EV3F).

6. **Figure 6D**: To substantiate the effect of TAX1BP1 binding on P62 clearance, autophagic flux assays are needed to confirm the turnover of P62. Additionally, this should be validated by depleting ATG5 and ATG14 alongside PI3K inhibitors.

We thank the reviewer for the suggestion. To assess p62 condensate turnover, we focused on NBR1, as its dynamic range is much higher in western blot analyses than of p62. We compared its protein levels in lysates from cell lines expressing WT and mutant TAX1BP1 induced by doxycycline. We found that the cells re-expressing WT TAX1BP1 showed a significant reduction in NBR1 protein levels in contrast to the mutants (see Fig. 6G). In addition, we would like to refer to the data shown in Fig. S1C, which show that TAX1BP1 accumulates upon blockage of autophagy at various stages.

7. **Figure 5**: The authors need to clarify the functional role underlying the competition between GABARAP and TAX1BP1 for NBR1 binding.

We are happy to address this point. We found that in our condensation assay, increasing the ubiquitin levels on the cargo (GST-8xUB vs GST-4xUB) rescues the recruitment of TAX1BP1 to the p62 condensates. This suggested to us, that upon reaching a certain local ubiquitin threshold at the condensates, TAX1BP1 can overcome the inhibitory effect of the GABARAP proteins and is then recruited to the condensates to induce autophagosome biogenesis by recruiting several upstream autophagy components such as FIP200 and TBK1 (see Fig. 7J, K).

8. **Figure S4F**: The gel should be replaced with a high-quality blot to ensure clarity and accuracy.

We agree and have repeated the Western blot (see Fig. EV4F).

Dear Sascha,

Thank you submitting a revised version of your manuscript. It was sent to the same reviewers that originally appraised your work; we received replies from two of them. I have attached their comments to the bottom of this email. As you will see, both are satisfied with the changes you made.

Before we can move forwards towards publication of your manuscript, though, there are some remaining editorial points which need to be addressed. In this regard, would you please:

- include the "Funding" section in the "Acknowledgments",
- change the 'Conflict of Interests' statement to the 'Disclosure and competing interests statement',
- remove the author credit section from the manuscript,
- check callouts for Fig. S1A, B, no such figure files have been uploaded,
- revise the author checklist; if the response is Yes, there should be section listed in the third (pink) column, but if the response is Not Applicable, the pink boxes should be blank. All "N.A." in the pink boxes should be removed. There is one example in the section "Data Availability" where the response is 'Not Applicable', but there is an explanation in the pink box: "Yes, link to the beads quantification tool was added in the methods section",
- remove instructions from the Reagents and Tools table,
- save Source Data files in a scheme of one figure/folder and then uploaded as .zip files. e.g. All the Source data files for figure 1 need to be saved in a single folder and this needs to be zipped and then uploaded as "SD figure 1.zip" file. For EV and/or appendix figures, ZIP together all source data,
- check that six different images have been used in the phase contrast panels of figure 5C,
- state in the legend of Figure 7D that images have been re-used from Figure 6A,
- provide exact p values are not provided in the legends of figures 1c-d, f; 2b, d, f; 3b-c, e, i-j; 4b, g; 5b, f; 6b-c, e-g, i-j; 7c, e, g-h, k; EV 2a; EV 3b; EV 6b, e; EV 7b,
- correct mismatch between annotated p values in the figure legend and the annotated p values in the figure file of figures 2d; 3b-c, e, i-j,
- correct the scale bar unit from μM to μm in figures 6h; EV 1c; EV 2b; EV 3c-e; EV 6c-d; (in the figure legend),
- define white arrows in the legends of figure 1b; 2e; 3d; 6a; 7d; EV 2b; EV 3a, c-d, and
- use a maximum of 5 EV figures. The extra figures should be compiled in Appendix PDF with the nomenclature Appendix Figure Sx and corresponding callouts. Appendix file needs to be in PDF format, and there should be a ToC with page numbers on the title page.

I look forward to receiving these changes. EMBO Press is an editorially independent publishing platform for the development of EMBO scientific publications.

Best wishes,

William

William Teale, PhD
Editor
The EMBO Journal
w.teale@embojournal.org

- a point-by-point response to the referees' comments, with a detailed description of the changes made (as a word file).
- a word file of the manuscript text.
- individual production quality figure files (one file per figure)
- a complete author checklist, which you can download from our author guidelines

(<https://www.embopress.org/page/journal/14602075/authorguide>).

- Expanded View files (replacing Supplementary Information)

- a Reagents and Tools Table as part of the Methods section, which can be downloaded from our author guidelines

(<https://www.embopress.org/page/journal/14602075/authorguide#structuredmethods>)

We realize that it is difficult to revise to a specific deadline. In the interest of protecting the conceptual advance provided by the work, we recommend a revision within 3 months (26th Dec 2024). Please discuss the revision progress ahead of this time with the editor if you require more time to complete the revisions. Use the link below to submit your revision:

Referee #1:

The authors addressed appropriately to the first review comments, and the revised manuscript is suitable to publication in EMBO J.

Referee #3:

Accept

All editorial and formatting issues were resolved by the authors.

Dear Sascha,

I am pleased to inform you that your manuscript has been accepted for publication in the EMBO Journal.

Congratulations to you and your team! I'm really looking forward to seeing this work in The EMBO Journal.

Yours sincerely,

William

William Teale, PhD
Editor
The EMBO Journal
w.teale@embojournal.org
